# Single-cell multiomics decodes regulatory programs for mouse secondary palate development

Fangfang Yan [1], Akiko Suzuki [2,3,4,7], Chihiro Iwaya [2,3,7], Guangsheng Pei [1], Xian Chen[1], Hiroki Yoshioka[2,3], Meifang Yu[1], Lukas M. Simon [5] ✉, Junichi Iwata [2,3,6] ✉ & Zhongming Zhao [1] ✉

Perturbations in gene regulation during palatogenesis can lead to cleft palate, which is among the most common congenital birth defects. Here, we perform single-cell multiome sequencing and profile chromatin accessibility and gene expression simultaneously within the same cells ($n$ = 36,154) isolated from mouse secondary palate across embryonic days (E) 12.5, E13.5, E14.0, and E14.5. We construct five trajectories representing continuous differentiation of cranial neural crest-derived multipotent cells into distinct lineages. By linking open chromatin signals to gene expression changes, we characterize the underlying lineage-determining transcription factors. In silico perturbation analysis identifies transcription factors SHOX2 and MEOX2 as important regulators of the development of the anterior and posterior palate, respectively. In conclusion, our study charts epigenetic and transcriptional dynamics in palatogenesis, serving as a valuable resource for further cleft palate research.

The development of the secondary palate is a dynamic, complex, and highly orchestrated three-dimensional process, involving growth, horizontal elevation, and fusion of the palatal shelves[1]. It is initiated around embryonic day (E) 12.5 in mice, when the palatal shelves arise from the lateral side of maxillary processes[2]. The palatal shelves grow vertically downward along with the tongue around E12.5 to E13.5 and elevate to the horizontal position at E14.0. Following the elevation, the bilateral palatal shelves grow toward the midline and fuse into the intact palate between E14.5 and E16.5[1,2].

Palatogenesis is regulated by the precise control of gene expression where transcription factors (TFs) bind to the accessible regulatory elements of the target genes[3]. Perturbations in palatogenesis may lead to cleft palate (CP), one of the most common congenital birth defects that has a significant long-term impact on patients' life quality[4]. Numerous efforts have aimed to uncover and annotate CP-associated genes[5,6], as well as the underlying regulatory mechanisms by integrating gene and microRNA expression data profiled from bulk tissue[7–9]. Beyond the bulk tissue level, single-cell RNA-sequencing (scRNA-seq) technologies have been applied to the soft palate[10] and fusing upper lip[11] and revealed large heterogeneity within the tissue and unique gene expression profile for each cell type.

While scRNA-seq reveals the transcriptional state differences between cells with high resolution, yet it provides little insight into the upstream regulations that drive such change[12]. Single-cell epigenome assays, such as single-cell assay for transposase-accessible chromatin with sequencing (scATAC-seq), capture open chromatin signals and decipher the regulation status[13]. Computational approaches have been developed to integrate independent scRNA-seq and scATAC-seq datasets of the same tissue[14]. However, inferring "anchors" between datasets may not fully recapitulate the true molecular processes[12].

[1]Center for Precision Health, School of Biomedical Informatics, The University of Texas Health Science Center at Houston, Houston, TX 77030, USA. [2]Department of Diagnostic and Biomedical Sciences, School of Dentistry, The University of Texas Health Science Center at Houston, Houston, TX 77054, USA. [3]Center for Craniofacial Research, The University of Texas Health Science Center at Houston, Houston, TX 77054, USA. [4]Department of Oral and Craniofacial Sciences, School of Dentistry, University of Missouri - Kansas City, Kansas City, Missouri 64108, USA. [5]Therapeutic Innovation Center, Baylor College of Medicine, Houston, TX 77030, USA. [6]MD Anderson Cancer Center UTHealth Graduate School of Biomedical Sciences, Houston, TX 77030, USA. [7]These authors contributed equally: Akiko Suzuki, Chihiro Iwaya. ✉e-mail: lukas.simon@bcm.edu; junichi.iwata@uth.tmc.edu; zhongming.zhao@uth.tmc.edu

More recently, single-cell multiomics technologies have emerged as a powerful approach to accurately deciphering regulation status. They profile gene expression and chromatin accessibility simultaneously within the same cells. Epigenetic changes at the DNA level can be directly linked to transcriptomic changes at the RNA level to reveal the interplay between regulatory DNA elements and the expression of target genes. Several multiomics technologies, such as sci-CAR[15], Paired-seq[16], SNARE-seq[12], and scCAT-seq[17], have been developed and applied to different tissue types, including human cerebral cortex[18], human lung cancer[17], and mouse kidney[15]. However, there is a lack of a comprehensive multiomics map of gene expression and its regulation during mouse secondary palate development at single-cell resolution.

In this study, we performed the single-cell multiome sequencing using 10x Multiome platform and profiled the transcriptome and epigenome simultaneously within the same cells isolated from the developing mouse secondary palate spanning four critical developmental stages. A total of eight major cell types were identified, which were defined by canonical marker gene expression. By mapping open chromatin signals to gene expression changes, we discovered cell-type-specific regulators with both enriched motif accessibility and gene expression. We then focused on cranial neural crest (CNC)-derived multipotent cells, reconstructed five developmental trajectories, and uncovered lineage-determining TFs that control the differentiation of each trajectory. This work reports a single-cell multiomic atlas of the developing mouse secondary palate. Insights into transcriptome and epigenome changes will increase our understanding of the underlying molecular processes, which provides a valuable resource for the community.

## Results

### Single-cell multiome dissects palate development

To dissect gene regulation mechanisms at the cellular level in the developing mouse secondary palate, we performed single-cell multiome sequencing using the 10x Chromium Single Cell Multiome platform. Following major developmental milestones of the mouse secondary palate as defined by the FaceBase consortium[19], we generated scRNA-seq and scATAC-seq libraries from the same cells at E12.5 ($n=2$), E13.5 ($n=3$), E14.0 ($n=2$), and E14.5 ($n=2$) (Fig. 1a). Jointly applying filters on both assays resulted in 36,154 cells with high-quality measurements across 32,285 genes and 123,807 accessible peaks representing potential regulatory elements (Supplementary Fig. 1a and Supplementary Table 1). The majority of cells had both high transcriptional start site (TSS) enrichment scores and a large number of fragments (Supplementary Fig. 1b). In addition, we observed nucleosome binding patterns (Supplementary Fig. 1c, left) and enrichment of chromatin accessibility around TSS when compared to the flanking regions (Supplementary Fig. 1c, right). Together, these results indicate high-quality scATAC-seq data.

Unsupervised dimension reduction based on either gene expression (scRNA-seq), chromatin accessibility (scATAC-seq) profiles, or both modalities combined, consistently revealed similar structures as visualized using Uniform Manifold Approximation and Projection (UMAP) (Fig. 1b and Supplementary Fig. 2a). Furthermore, biological replicates from the same developmental stage showed strong overlap in the UMAP embedding, indicating high-quality data (Supplementary Fig. 1d).

Next, we performed clustering analysis followed by cell type annotation. Compared to scRNA-seq, there is limited knowledge of cell-type-specific chromatin accessibility. Therefore, cell type annotation is more challenging in scATAC-seq data[20]. One frequently used computational approach involves cross-modality integration and label transfer from reference scRNA-seq data[21]. Here, single-cell multiomics technologies eliminate the need for inferring relationships in silico, but instead allow direct annotation of scATAC-seq-based clusters using cell type labels derived from the scRNA-seq data. We observed distinct clusters in both RNA and ATAC data, representing eight major cell types (Fig. 1c, d). The annotation results were robust across various cutoffs for the number of highly variable genes (Supplementary Fig. 2b). Each cell type was defined by canonical marker gene expressions, including CNC-derived mesenchymal cells (*Prrx1*, $n=28,529$, 78.91%), epithelial cells (*Krt14*, $n=5866$, 16.23%), endothelial (*Cdh5, Cldn5*, $n=714$, 1.97%), myeloid (*Lyz2*, $n=397$, 1.10%), glial cells (*Plp1, Sox10*, $n=307$, 0.85%), myogenic precursors (*Myf5*, $n=200$, 0.55%), neuronal cells (*Tubb3, Stmn2*, $n=113$, 0.31%) and myocytes (*Myh7*, $n=28$, 0.08%) (Fig. 1d, Supplementary Fig. 3a). We quantified the gene activity score, a metric defined by aggregating accessible chromatin regions intersecting the gene body and promoters in scATAC-seq data. The above-mentioned markers, which showed cell type-specific expression, also exhibited similar patterns of chromatin accessibility in corresponding clusters (Fig. 1e).

To further corroborate our inferred cell type identities, we analyzed a recently published scRNA-seq dataset of the developing mouse soft palate[10]. Following normalization and dimension reduction, we projected our scRNA-seq data into the published soft palate manifold using canonical correlation analysis[14] (see "Methods") and observed high agreement between cell type annotations (Supplementary Fig. 3b, c).

### Cell type-specific multiomic markers

Paired RNA and ATAC measurements from the same cell reveal both the transcriptional state and the upstream DNA regulatory element activities, which allows direct mapping of epigenetic gene regulation to gene expression. We aimed to identify TFs with both enriched accessibility and expression profiles for a specific cell type, representing putative markers for that cell type.

To this end, we first conducted differential gene expression analysis between cell types and identified 6573 differentially expressed genes (DEGs) (adjusted $P$ value < 0.05 and log Fold change >0.1). To find regulatory elements for each DEG, we conducted peak-gene linkage analysis by calculating the Pearson correlation coefficient (PCC) between chromatin accessibility and gene expression while accounting for peak size and fragment count. Positively linked peak-gene pairs may represent enhancer-gene interactions. A total of 15,018 pairs were identified, including 12,596 regulatory elements significantly linked to 3787 cell type-specific genes (Fig. 2a, correlation >0, adjusted $P$ value < 0.05). Each cell-type-specific gene was linked to a median of three peaks (min = 1, max = 28, mean = 3.966).

As an example, in CNC-derived mesenchymal cells, the locus at chr12: 33,957,146–33,958,061 was mapped to the promoter region of *Twist1* and showed the most significant association (correlation = 0.315, adjusted $P$ value = $1.70 \times 10^{-8}$). Both the expression level of *Twist1* and accessibility of chr12: 33,957,146–33,958,061 were increased in CNC-derived mesenchymal cells compared to all other cell types (Fig. 2b). Genome browser visualization of the *Twist1* locus revealed that chr12: 33,957,146–33,958,061 partially overlaps the transcription start site (TSS) of *Twist1*. Therefore, chr12: 33,957,146–33,958,061 most likely acts as an enhancer that upregulates the expression of *Twist1* in CNC-derived mesenchymal cells (Fig. 2b). Another exemplary peak-gene pair was *Tie1* and locus chr4: 118,489,480–118,490,171, showing significant enrichment in endothelial cells (correlation = 0.369, adjusted $P$ value = $3.88 \times 10^{-14}$, Supplementary Fig. 4).

To nominate TFs that control each major cell type, we identified enriched motifs of these peaks. The following criteria were then applied to define cell-type-specific TFs: (1) TF expression was enriched at the RNA level, and (2) TF binding motif accessibility was enriched in the ATAC measurements. In total, we discovered 81 putative markers for the eight major cell types (adjusted RNA $P$ value < 0.05 and adjusted motif $P$ value < 0.05, Table 1). For example, in CNC-derived mesenchymal cells, both the RNA expression of *Twist2* (*RNA P*

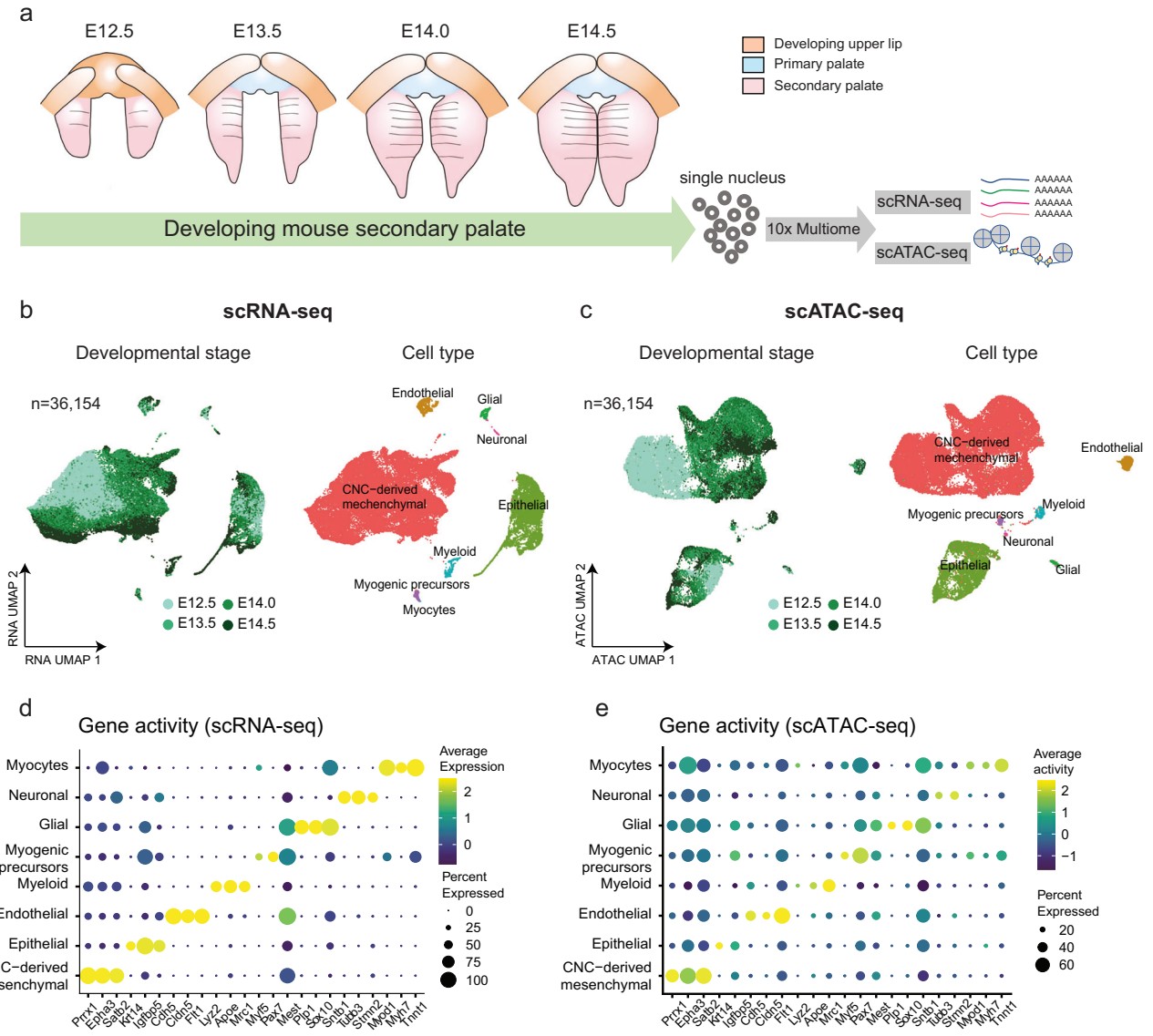

**Fig. 1 | Single-cell multiome assays dissect transcriptome and epigenome changes of the developing mouse secondary palate. a** Schematic plot depicts the development of mouse secondary palate (pink) spanning embryonic day (E) 12.5 ($n = 2$), E13.5 ($n = 3$), E14.0 ($n = 2$), and E14.5 ($n = 2$). The isolated nuclei were subjected to 10x Chromium Multiome sequencing to profile scRNA-seq and scATAC-seq simultaneously within the same cells. **b, c** Uniform manifold approximation and projection (UMAP) visualization of 36,154 cells based on RNA assay (**b**) or ATAC assay (**c**). Each dot represents one cell and is colored by the developmental stage (left) and annotated cell types (right). **d, e** Dot plot illustrates marker gene expression (*x* axis) (**d**) and gene activity (**e**) (*x* axis) across cell types (*y* axis). Dot size is proportional to the percent of expressed cells. Colors indicate low (purple) to high (yellow) expression.

value = $3.00 \times 10^{-241}$) and chromatin accessibility of the *Twist2* binding motif MA0633.1 (*motif P* value = 0) were significantly enriched (Fig. 2c). In agreement with our data, previous studies show that mice deficient for both *Twist1* and *Twist2* exhibit CP[22,23].

**CNC-derived subtypes reflect in vivo anatomical locations**
Among the cell types in the developing palate, CNC-derived mesenchymal cells are the most abundant and are considered an important cell lineage for palate development[24]. To understand the heterogeneity within CNC-derived mesenchymal cells, we isolated this cluster and conducted an independent analysis, including normalization, clustering, and dimension reduction (Fig. 3a, left panel). In the dimension-reduced data manifold, we observed a continuous progression from E12.5 to E14.5 (Fig. 3a and Supplementary Fig. 5a). Cells from the early stage (E12.5) were more homogeneous compared to cells from later stages and accordingly resided near the center of the low-dimensional manifold. Cells from the late stage (E14.5), on the

other hand, resided at the edge of the manifold, most likely representing more differentiated cell states.

Cell subtype annotation was conducted through extensive manual curation of marker genes. Notably, we identified seven subtypes characterized by specific gene expression signatures (Fig. 3a, right panels). For example, anterior palatal mesenchymal cells exhibited high expression of ALX Homeobox 1 (*Alx1*) (Supplementary Fig. 5b). Chondrogenic cells were characterized by high expression of *Sox9* and *Col12a1* while osteoblasts had high expression of *Runx2* and *Sp7*. Dental mesenchymal cells exhibited high expression of *Dlx2*, *Sostdc1*, and *Tfap2b*. Posterior palatal mesenchymal cells had high expression of *Tbx22*. Progenitor-related genes were highly expressed in cluster 5, such as *Dach1*, *Lmo4*, *Hmgb2*, *Hmgb3*, and *Runx1t1* (adjusted *P* values < $2.2 \times 10^{-16}$). Gene Set Enrichment Analysis (GSEA) revealed that these genes were significantly associated with the regulation of stem cell proliferation [GO: 0072091, false discovery rate (FDR) = $2.58 \times 10^{-3}$, enrichment ratio = 92.87] and enriched in neural progenitor cells

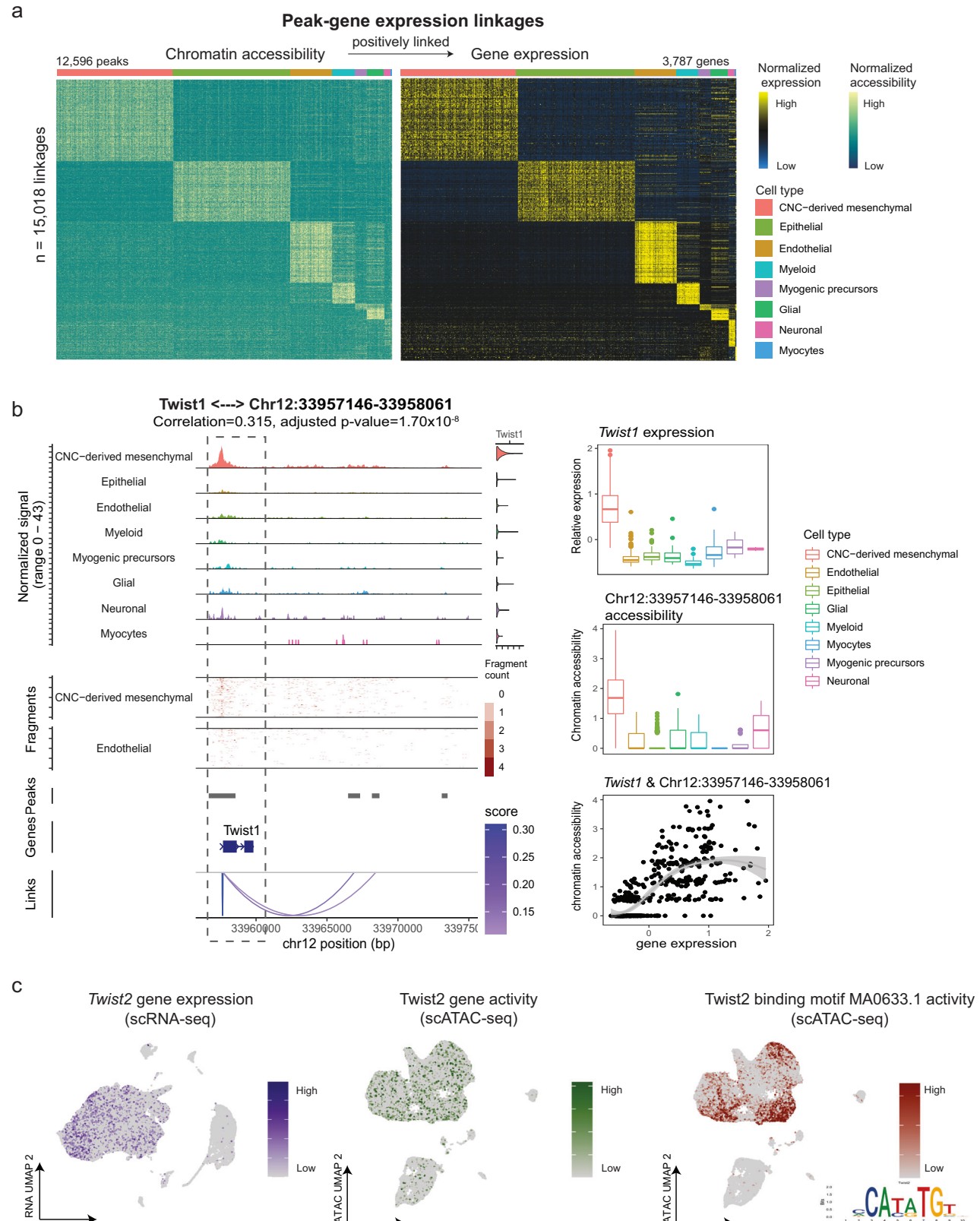

**a** Peak-gene expression linkages

**b** Twist1 <---> Chr12:33957146-33958061
Correlation=0.315, adjusted p-value=1.70x10⁻⁸

**c** *Twist2* gene expression (scRNA-seq) — Twist2 gene activity (scATAC-seq) — Twist2 binding motif MA0633.1 activity (scATAC-seq)

(FDR = $2.68 \times 10^{-3}$, enrichment ratio = 13.97) (Supplementary Fig. 5c), suggesting cluster 5 represented CNC-derived multipotent cells.

Of note, gene expression patterns aligned with in vivo anatomical locations (Fig. 3b). For example, *Shox2* and *Msx1* were specifically expressed in anterior regions. In contrast, expression of *Meox2* and *Tbx22* was restricted to posterior regions[25]. Even though expression

patterns of *Shox2* and *Meox2* have been studied, the majority of region-specific genes we identified in the developing palate in this study have not been well characterized yet (Table 2).

To validate the gene signatures of anterior and posterior sub-populations, we performed bulk RNA-sequencing (RNA-seq) after isolating RNA from the microdissected anterior 1/3 ($n = 3$) and

**Fig. 2 | Integration of chromatin accessibility with gene expression reveals cell-type-specific multiomic marker genes. a** Heatmap shows normalized chromatin accessibility (left) and gene expression (right) of 15,018 significantly linked peak-gene pairs. Each row represents a positively linked pair of regulatory elements and genes. Bar on the top represents major cell types. **b** Left: Genome Browser visualization of aggregated chromatic accessibility at the chr12: 33.96–33.97 (Mb) locus for each major cell type, coupled with *Twist1* gene expression. Arcs at the bottom denote positively linked peak-*Twist1* pairs. The linkage between *Twist1* and chr12: 33,957,146–33,958,061 is highlighted with gray dotted box. The Pearson correlation test is employed for each pair of peak-gene, and multiple test adjustments are applied. Right: Boxplots show the expression level of *Twist1* (top) and accessibility of the linked peak chr12: 33,957,146–33,958,061 (middle) across cell types. Minima:

Lower limit of the whisker. Maxima: Upper limit of the whisker. Centre: Median line inside the box. The upper and lower box bounds represent the 25% and 75% percentile of data. The scatter plot shows the significant correlation between *Twist1* expression and chr12: 33,957,146–33,958,061 accessibility (bottom). Each dot represents the aggregated gene expression and accessibility of ten cells from $n = 28,529$ biologically independent CNC-derived mesenchymal cells. The shaded areas around the fitted smooth line represent the error bands, indicating the 95% confidence interval. **c** UMAP visualizations illustrate the multimodal profiling of *Twist2,* including gene expression (left), gene activity (middle), and TWIST2 binding motif MA0633.1 activity (right). The position weight matrix of the MA0633.1 motif is embedded in the bottom right corner.

posterior 1/3 regions ($n = 3$) of the developing secondary palate at E14.0. The expression patterns in the bulk RNA-seq data confirmed our scRNA-seq results. The majority of genes (75/86, 87.21%) that were upregulated in the scRNA-seq anterior cluster were also upregulated in bulk RNA-seq from anterior tissue (Fig. 3c). Fisher's Exact test revealed significant enrichment between scRNA-seq DEGs and bulk RNA-seq DEGs (odds ratio = 1354.47, $P$ value $< 2.2 \times 10^{-16}$). For example, *Shox2*

### Table 1 | Top five putative regulators in each major cell type

| Cell type | Gene symbol | RNA. *P* value | Motif | Motif. *P* value |
|---|---|---|---|---|
| CNC-derived mesenchymal | *Twist2* | 3.00E-241 | MA0633.1 | 0.00E + 00 |
| | *Lhx8* | 0.00E + 00 | MA0705.1 | 2.01E-61 |
| | *Foxf1* | 2.97E-192 | MA1606.1 | 5.41E-133 |
| | *Shox2* | 6.76E-211 | MA0720.1 | 1.72E-73 |
| | *Tcf3* | 2.01E-05 | MA0092.1 | 2.03E-257 |
| Epithelial | *Sox2* | 0.00E + 00 | MA0142.1 | 2.94E-88 |
| | *Sox6* | 2.16E-225 | MA0515.1 | 2.05E-197 |
| | *Smad3* | 1.38E-144 | MA1622.1 | 8.26E-215 |
| | *Klf12* | 1.00E-09 | MA0742.1 | 0.00E + 00 |
| | *Runx1* | 2.58E-233 | MA0002.2 | 1.30E-36 |
| Endothelial | *Foxo1* | 8.90E-145 | MA0480.1 | 4.91E-18 |
| | *Nr2f6* | 7.22E-07 | MA0677.1 | 3.08E-113 |
| | *Sox17* | 0.00E + 00 | MA0078.1 | 9.56E-30 |
| | *Nr5a2* | 7.58E-145 | MA0505.1 | 5.63E-37 |
| | *Esrrg* | 1.69E-15 | MA0643.1 | 1.39E-57 |
| Myeloid | *Runx1* | 6.16E-104 | MA0002.2 | 1.24E-08 |
| | *Nfe2l2* | 1.67E-65 | MA0150.2 | 5.29E-29 |
| | *Arnt* | 4.93E-04 | MA0004.1 | 9.49E-21 |
| | *Bach1* | 7.34E-03 | MA0591.1 | 3.27E-23 |
| | *Nr4a2* | 1.72E-03 | MA0160.1 | 5.49E-22 |
| Myogenic precursors | *Tcf12* | 2.09E-06 | MA0521.1 | 1.73E-115 |
| | *Tcf21* | 3.08E-68 | MA0832.1 | 5.97E-105 |
| | *Plagl1* | 2.07E-16 | MA1615.1 | 1.68E-03 |
| | *Arx* | 3.20E-63 | MA0874.1 | 7.88E-06 |
| | *Nobox* | 3.39E-04 | MA0125.1 | 4.40E-06 |
| Glial | *Sox5* | 8.83E-146 | MA0087.1 | 1.65E-04 |
| | *Nr4a2* | 5.35E-48 | MA0160.1 | 2.39E-42 |
| | *Creb5* | 1.91E-63 | MA0840.1 | 3.30E-22 |
| | *Dlx1* | 1.64E-138 | MA0879.1 | 1.23E-04 |
| | *Rxra* | 3.50E-04 | MA0065.2 | 3.97E-52 |
| Neuronal | *Arid3b* | 1.37E-06 | MA0601.1 | 3.29E-05 |
| | *Hmx2* | 5.68E-177 | MA0897.1 | 8.05E-03 |
| | *Zic1* | 1.03E-06 | MA1628.1 | 8.97E-03 |

The Wilcoxon rank-sum test is utilized, and the Benjamini–Hochberg method is applied for multiple test adjustment of the *P* value.

and *Msx1* were significantly higher expressed in the anterior region (bulk RNA-seq adjusted $P$ value $= 9.12 \times 10^{-49}$, scRNA-seq adjusted $P$ value $< 2.20 \times 10^{-16}$), while *Meox2* and *Tbx22* exhibited higher expression in the posterior region (bulk RNA-seq adjusted $P$ value $= 1.69 \times 10^{-94}$, scRNA-seq $P$ value $< 2.20 \times 10^{-16}$, Fig. 3d). Although the highly significant correspondence between bulk and single-cell levels, several differentially expressed genes with large fold changes in bulk RNA-seq were not detected in the scRNA-seq data. These genes tended to be expressed at low levels in bulk RNA-seq, suggesting that the limited sensitivity of lowly expressed transcripts in the scRNA-seq assay may obscure the signals (Supplementary Fig. 6).

To further validate these results, the top five region-specific genes for the anterior (*Shox2, Satb2, Inhba, Cyp26b1,* and *Nrp1*) and the posterior subpopulations (*Meox2, Prickle1, Sim2, Efnb2,* and *Trps1*) were analyzed with quantitative reverse-transcription polymerase chain reaction (qRT-PCR) using E14.0 anterior and posterior palatal shelves (Fig. 3e). Furthermore, RNAscope in situ hybridization confirmed expression patterns of these genes (Fig. 3f). *Shox2, Satb2,* and *Nrp1* were expressed mainly in the anterior and middle regions of the palate, but not in the posterior palate. The expression of *Cyp26b1* was restricted to the anterior region and *Inhba* was restricted to beneath the epithelial layer in the anterior and middle region. In contrast, *Meox2, Prickle1,* and *Efnb2* were mainly expressed in the posterior region of the palate. Interestingly, *Sim2* and *Trps1* were expressed medially in the anterior half of the posterior region of the developing secondary palate. Overall, these results validated our subtype annotations of CNC-derived mesenchymal cells which corresponded to anatomical locations.

### Cell fate analysis unveils lineage-determining regulators

Single-cell data from discrete time points can be considered "snapshots" of the underlying continuous developmental process[26]. To connect static snapshots into a "movie" and computationally reconstruct the molecular dynamics during the differentiation of CNC-derived mesenchymal cells, we applied Wadding-Optimal Transport (WOT)[27,28], an algorithm designed for trajectory analysis of time series scRNA-seq data. WOT connects adjacent time points by finding the most probable cell transition paths using the mathematical theory of optimal transport[29]. The resulting trajectories can be used to model cell fate decisions with high resolution.

As chondrocytes originate from the pterygoid plate of the sphenoid bone and are not considered as part of the secondary palate[10], we excluded them from downstream analysis (Fig. 4a). Simulated random walks based on the WOT-derived cell transition matrix showed that most trajectories started from CNC-derived multipotent cells (black dots) and terminated at various subpopulations at later stages (yellow dots) (Fig. 4b, left panel). We then quantified the terminal state likelihood of each cell and defined those with high likelihoods as terminal state cells (Fig. 4b, right panel). Next, we computed the probabilities that an early cell would transition towards any terminal cell states. Overall, we discovered five distinct trajectories, representing the continuous differentiation of multipotent cells into (1) anterior palatal

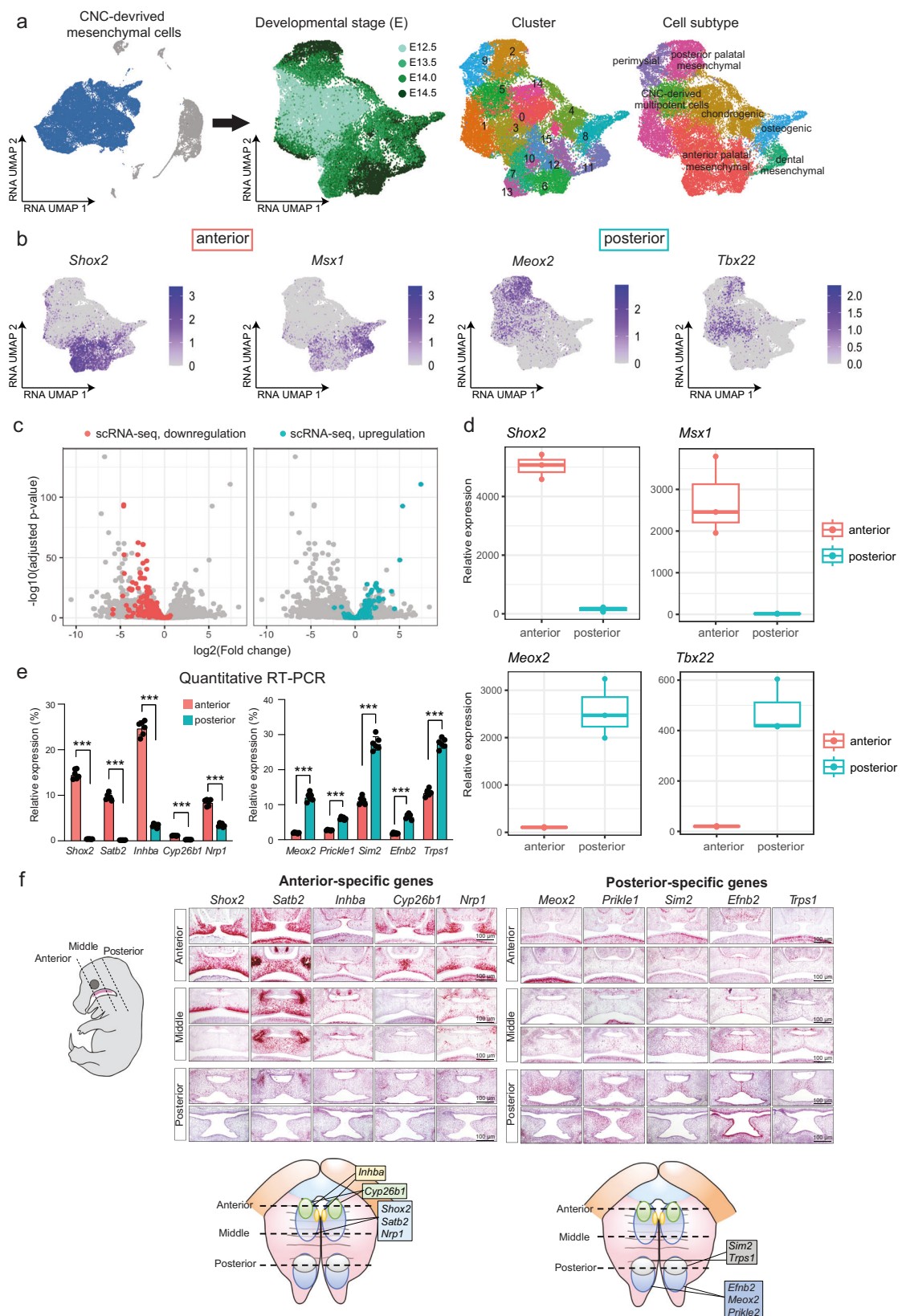

mesenchymal cells, (2) posterior palatal mesenchymal cells, (3) dental mesenchymal cells, (4) osteoblasts, and (5) perimysial cells (Fig. 4c and Supplementary Fig. 7). Next, we calculated RNA velocity, which predicts future states of individual cells based on the stratification of spliced and unspliced mRNAs[30]. As shown in Fig. 4d, the directed dynamic information based on RNA velocity was consistent with

inferred trajectories. The RNA velocities pointed away from the cells at E12.5 in the middle of the embedding towards later time points at the border of the embedding (Fig. 4d).

To more granularly resolve these velocity predictions, we applied CellRank, which infers initial and terminal state probabilities for each cell based on RNA velocity[28]. Consistent with WOT-derived

**Fig. 3 | Anterior and posterior subpopulations were identified and characterized. a** Left: UMAP visualization of the whole dataset with CNC-derived mesenchymal cells highlighted in blue. Right: Independent UMAP visualizations of CNC-derived mesenchymal cells colored by developmental stage, cluster, and cell subtype, respectively. **b** Feature plot shows the expression of representative genes in the anterior and posterior, respectively. **c** Volcano plot shows the log2 Fold Change and negative log10 of adjusted *P* values for each gene in the bulk RNA-seq dataset and colored by significance in the single-cell RNA-seq dataset. The Wald test is employed as a two-sided test, and multiple test adjustments are applied. **d** Boxplot shows the expression of four representative DEGs, *Shox2, Msx1, Meox2*, and *Tbx22* at the bulk level. *n* = 6 biologically independent samples. Minima: Lower limit of the whisker. Maxima: Upper limit of the whisker. Centre: Median line inside the box. The upper and lower box bounds represent the 25% and 75% percentile of data. **e** Bar plot shows quantitative RT-PCR of genes in anterior (red bars) and posterior (blue bars) regions. ***$P < 0.001$. $n = 6$ independent experiments. Data are presented as mean values +/− standard error of the mean (SEM). *T* test, two-sided. The anterior-specific genes include *Shox2* (adjusted *P* value = $1.10 \times 10^{-11}$), *Satb2* (adjusted *P* value = $2.27 \times 10^{-11}$), *Inhba* (adjusted *P* value = $1.69 \times 10^{-11}$), *Cyp26b1* (adjusted *P* value = $3.95 \times 10^{-12}$), and *Nrp1* (adjusted *P* value = $3.05 \times 10^{-8}$). The posterior-specific genes comprise *Meox2* (adjusted *P* value = $1.29 \times 10^{-9}$), *Prickle1* (adjusted *P* value = $2.19 \times 10^{-9}$), *Sim2* (adjusted *P* value = $6.05 \times 10^{-9}$), *Efnb2* (adjusted *P* value = $2.94 \times 10^{-8}$), and *Trps1* (adjusted *P* value = $1.84 \times 10^{-9}$). Source data are provided as a Source Data file. **f** RNAscope in situ hybridization validated gene expression signature in anterior and posterior subpopulations. Left: microscope images (100 μm) show expression patterns for each gene by in situ hybridization. Right: anatomical mouse embryo images outline gene expression patterns. This validation process was conducted in six independent experiments, each yielding consistent results.

trajectories, CellRank found high initial state probabilities in CNC-derived multipotent cells and high terminal state probabilities in the late-stage subpopulations (Supplementary Fig. 8). To analyze cell fate decisions within the progenitor populations within the progenitor populations, we isolated those with high transition probabilities toward the anterior and posterior trajectories, respectively.

## Table 2 | Marker genes in anterior and posterior subpopulations

| Subpopulation | Gene symbol | Average log2(fold change) | Adjusted *P* value | Cleft-related gene |
|---|---|---|---|---|
| Anterior palatal mesenchymal subpopulation | *Shox2* | 2.48 | 2.84E-124 | Yes |
| | *Satb2* | 1.74 | 2.49E-105 | No |
| | *Zfhx4* | 1.66 | 4.80E-79 | No |
| | *Adgrl2* | 1.22 | 3.45E-73 | No |
| | *Alx1* | 1.21 | 1.20E-62 | Yes |
| | *2700069I18Rik* | 1.05 | 5.86E-56 | No |
| | *Eya1* | 1.24 | 2.00E-54 | Yes |
| | *Thsd4* | 1.27 | 6.39E-54 | No |
| | *Mme* | 1.23 | 6.44E-50 | No |
| | *Slit2* | 1.60 | 2.81E-47 | No |
| | *Sox5* | 1.28 | 7.45E-46 | Yes |
| | *Ror1* | 0.91 | 6.19E-40 | No |
| | *Asb4* | 1.13 | 6.69E-39 | No |
| | *Six1* | 0.86 | 9.30E-39 | Yes |
| | *Msx1* | 0.66 | 1.71E-34 | Yes |
| Posterior palatal mesenchymal subpopulation | *Meox2* | 1.67 | 1.10E-110 | Yes |
| | *Dach1* | 2.17 | 4.02E-107 | No |
| | *Pcdh9* | 2.45 | 2.59E-97 | No |
| | *Col25a1* | 2.20 | 6.60E-95 | No |
| | *Inpp4b* | 2.60 | 6.64E-90 | No |
| | *Prickle1* | 1.47 | 1.77E-85 | Yes |
| | *Sim2* | 1.23 | 3.38E-78 | Yes |
| | *Lmo4* | 1.73 | 4.50E-75 | No |
| | *Pcdh15* | 3.10 | 1.09E-73 | No |
| | *Cdh18* | 1.97 | 6.17E-68 | No |
| | *Hs3st5* | 1.49 | 2.58E-64 | No |
| | *Efnb2* | 1.15 | 7.93E-64 | Yes |
| | *Tmtc2* | 1.58 | 1.15E-61 | No |
| | *Fhod3* | 1.45 | 7.66E-61 | No |
| | *Pam* | 1.22 | 3.93E-50 | No |

The Wilcoxon rank-sum test is utilized, and the Benjamini–Hochberg method is applied for multiple test adjustment of the *P* value.

Differential gene expression analysis unveiled distinct expression profiles in these two subpopulations. In the progenitor anterior subpopulation, genes such as *Shox2, Satb2, Inhba, Cyp26b1*, and *Nrp1* demonstrated significantly higher expression levels. Conversely, in the progenitor posterior subpopulation, genes like *Meox2, Prickle1, Sim2, Efnb2*, and *Trps1* exhibited elevated expression (Supplementary Fig. 9). The results align with the expression profile observed in terminally differentiated anterior and posterior populations, as illustrated in Fig. 3. Taken together, these analyses generated high-resolution pseudotemporal trajectories representing the continuous expression changes that occur during the development of the secondary palate. Next, we investigated both gene expression and regulation dynamics of each trajectory. The cells with large diffusion pseudotime values tended to be derived from late time points (Supplementary Fig. 10).

To validate the accuracy of the inferred trajectories, we conducted a comprehensive analysis of previously published H3K27 acetylation data obtained from both the anterior and posterior palate[31]. Leveraging our extensive multiomic datasets, we identified accessibility peaks associated with the genes along the anterior and posterior developmental pathways, respectively. We then assessed the probability of observing enriched overlap between the scATAC anterior and posterior peaks with the corresponding anterior and posterior acetylation tracks. The results revealed statistically significant enrichment for the anterior trajectory (Fisher's Exact test, odds ratio = 1.50, *P* value = $4.81 \times 10^{-3}$, Supplementary Fig. 11). The posterior trajectory also revealed increased enriched overlap, albeit not reaching statistical significance (Fisher's Exact test, odds ratio = 1.34, *P* value = 0.13). Taken together, these data confirm the reliability of our inferred trajectories.

We first focused on the anterior palatal mesenchymal trajectory. To pinpoint the driver genes for this trajectory, we conducted association tests between the expression level of each gene and estimated fate probability to each terminal state. Those with significant positive correlations were defined as driver genes. We identified a total of 556 driver genes (correlation >0.05 and adjusted *P* value < 0.05) (Fig. 4e–h). The top hits included *Shox2* (correlation = 0.468, adjusted *P* value < $2.2 \times 10^{-16}$), *Foxd2os* (correlation = 0.391, adjusted *P* value < $2.2 \times 10^{-16}$), and *Foxd2* (correlation = 0.333, adjusted *P* value < $2.2 \times 10^{-16}$) (Supplementary Fig. 12a).

To further investigate when and how these driver genes were regulated along the trajectory, we extracted 7240 cells with high probabilities to differentiate towards the anterior trajectory (fate probability >75% quantile) and ordered them by diffusion pseudotime. We then performed peak-gene linkage analysis as described above and connected expression trajectories with chromatin accessibility dynamics. Out of 984 peak-gene linkages, 428 (43.49%) were significantly linked (correlation >0 and adjusted *P* value < 0.05). We observed consistent gene expression and chromatin accessibility dynamics along the anterior trajectory (Fig. 4e).

Using k-means clustering, these driver genes were divided into three groups, showing increased expression specifically at the start,

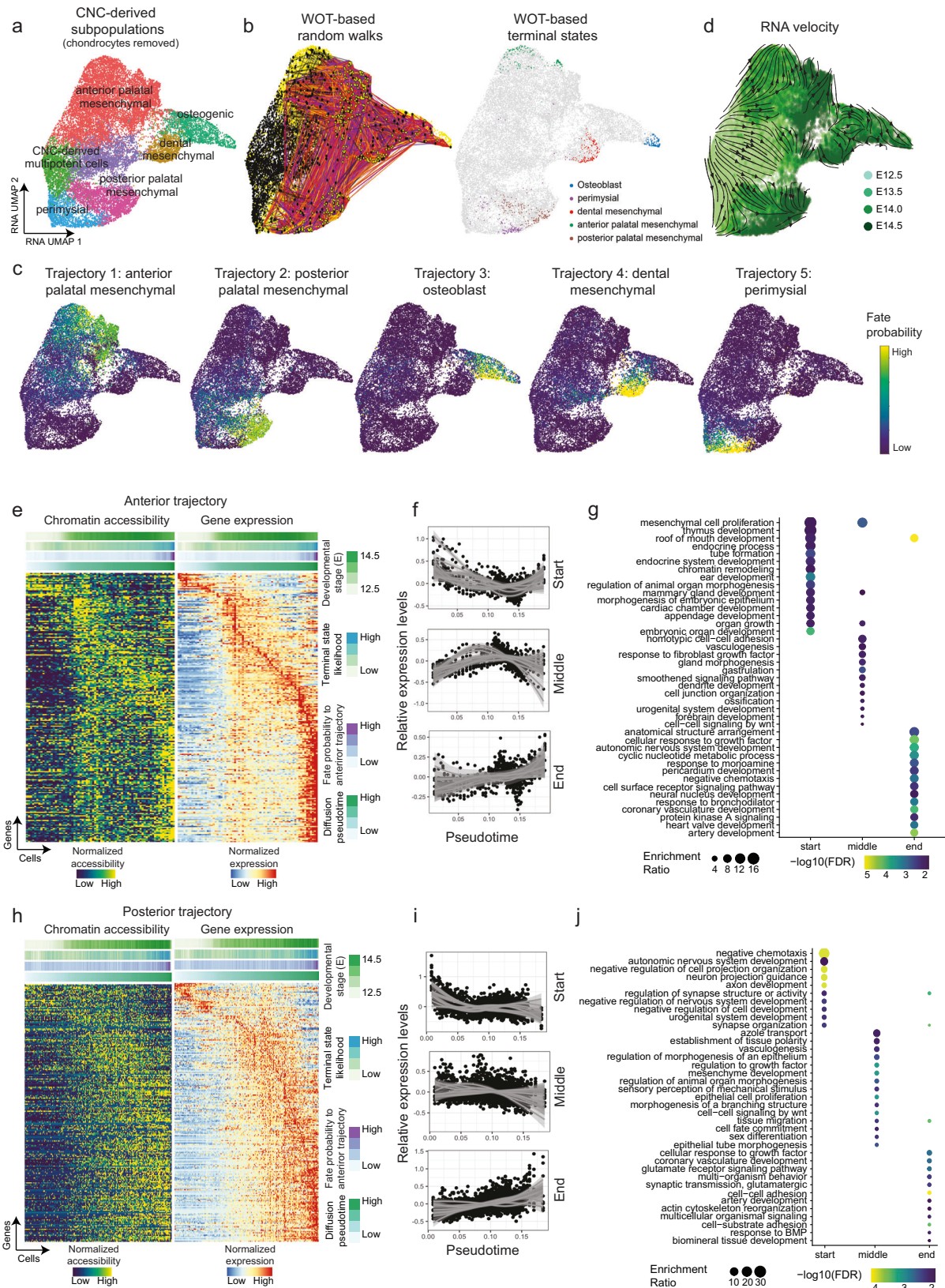

**e** Anterior trajectory

**f**

**g**

**h** Posterior trajectory

**i**

**j**

middle, or end of the anterior trajectory (Fig. 4f). The genes upregulated at the start of the anterior trajectory were enriched in mesenchymal cell proliferation (enrichment ratio = 16.71, adjusted $P$ value = 0.036), roof of mouth development (enrichment ratio = 12.73, adjusted $P$ value = 0.022), and chromatin remodeling (enrichment ratio = 9.90, adjusted $P$ value = 0.035) (Fig. 4g). The genes upregulated

at the middle of the trajectory were enriched in mesenchymal cell proliferation (enrichment ratio = 11.14, adjusted $P$ value = $2.28 \times 10^{-3}$) and response to fibroblast growth factor (enrichment ratio = 5.74, adjusted $P$ value = 0.027) while the genes upregulated at the end of the trajectory were associated with the roof of mouth development (enrichment ratio = 7.49, adjusted $P$ value = $1.04 \times 10^{-6}$) and

**Fig. 4 | Reconstruction of CNC-derived mesenchymal trajectories by Optimal Transport reveals lineage-determining transcription factors. a** UMAP visualization of CNC-derived mesenchymal subpopulations with chondrocytes removed. **b** UMAP visualization colored by (left) WOT-based random walks and (right) terminal states. The random walks are based on the WOT-derived cell–cell transition matrix. Black dots represent the start points of the trajectory, while yellow dots represent endpoints. **c** UMAP visualizations show fate probabilities to each trajectory. Cells are colored by probabilities (high: yellow, low: purple). **d** UMAP visualization with streamlines and arrows shows RNA velocity-derived information. Each point represents one cell and is colored by developmental stages. Streamlines and arrows represent the future directions for each cell. **e** Heatmap shows paired chromatin accessibility and gene expression (rows) for the anterior trajectory (columns). Each row represents a putative pair of genes and linked regulatory elements. Bars on the top represent diffusion pseudotime, fate probabilities to the anterior palatal mesenchymal trajectory, terminal state likelihood, and developmental stage. Columns are ordered by diffusion pseudotime. **f** Scatter plot with fitted lines shows the expression pattern for each group of driver genes along the trajectory. The shaded areas around the fitted smooth line represent the error bands, indicating the 95% confidence interval. **g** Dot plot shows enriched pathways (*y* axis) for each driver gene group (*x* axis) upregulated at the start, middle, and end of anterior palatal mesenchymal trajectory. Dot is scaled by enrichment ratio and colored by significance. **h–j** Similar visualizations to (**e–g**) for the posterior palatal mesenchymal trajectory.

anatomical structure arrangement (enrichment ratio = 10.49, adjusted *P* value = $4.58 \times 10^{-3}$).

We then applied an analogous approach to the posterior palatal mesenchymal trajectory and identified 586 driver genes, such as *Col25a1* (correlation = 0.518, adjusted *P* value < $2.2 \times 10^{-16}$), *Meox2* (correlation = 0.496, adjusted *P* value < $2.2 \times 10^{-16}$), and *Inpp4b* (correlation = 0.400, adjusted *P* value < $2.2 \times 10^{-16}$) (Supplementary Fig. 12b). Among them, 216 genes were significantly regulated by 353 regulatory elements (Fig. 4h). Pathway enrichment analysis of these genes revealed the involvement of neuron-related pathways in the early stage (enrichment ratio = 11.30, adjusted *P* value = $8.99 \times 10^{-5}$), mesenchymal development (enrichment ratio = 5.42, adjusted *P* value = $1.37 \times 10^{-4}$), and tissue migration (enrichment ratio = 3.57, adjusted *P* value = $4.09 \times 10^{-4}$) in the intermediate and late stages of the trajectory (Fig. 4i, j).

We next investigated underlying transcriptional regulators by motif enrichment analysis. We characterized a list of lineage-determining TFs that are potential candidates to control each trajectory by binding to regulatory elements to regulate the expression of the above-mentioned driver genes. Shox2 was identified as an important regulator at the start of the anterior trajectory (motif adjusted *P* value = $6.09 \times 10^{-3}$, motif fold change = 4.21, gene adjusted *P* value < $2.2 \times 10^{-16}$, gene fate correlation = 0.47) (Fig. 5a). MEOX2 showed potential regulatory roles in the middle (motif adjusted *P* value = 0.024, motif fold change = 2.87) and end of the posterior trajectory (motif adjusted *P* value = $1.85 \times 10^{-3}$, motif fold change = 2.06, Fig. 5b). To experimentally validate these predictions, we conducted chromatin immunoprecipitation followed by sequencing (ChIP-seq) experiments. SHOX2 and MEOX2 ChIP-seq data were generated from anterior and posterior palate tissue, respectively. We calculated the likelihood of observing a ChIP-seq peak near the genes that were upregulated at the start of the anterior trajectory and the middle of the posterior trajectory. As computationally predicted, the likelihood of observing a SHOX2 peak was increased for genes upregulated at the start of the anterior trajectory. Correspondingly, the likelihood of observing a MEOX2 peak was increased for genes upregulated at the middle of the posterior trajectory (Fig. 5c). Despite limited statistical power given the use of duplicates, these results showed marginal significance (One-sided *t* test, anterior start *P* value = 0.055, posterior middle *P* value = 0.09). For example, we observed a ChIP-seq binding peak for SHOX2 but not MEOX2 in the predicted SHOX2 target *Nfia* (*P* value = $2.29 \times 10^{-4}$, signal = 3.01, Fig. 5d). Simultaneously, a MEOX2 binding peak was observed upstream of the predicted MEOX2 target *Has2* (*P* value = $1.53 \times 10^{-4}$, signal = 2.82, Fig. 5e).

To further validate these predictions, we first examined the odds ratio distribution of fate probabilities towards anterior versus posterior trajectories. As expected, *Shox2*-positive cells had high probabilities to differentiate towards the anterior palatal mesenchymal trajectory (Supplementary Fig. 13a, left panel) while *Meox2*-positive cells had high probabilities to differentiate towards the posterior trajectory (Supplementary Fig. 13b, left panel). Those terminally differentiated cells at E14.5 emerged from the multipotent cells at the early stage (E12.5) (Supplementary Fig. 13a, b, right panels).

In addition, we observed a significant negative correlation of fate probabilities between the anterior and posterior trajectories (PCC Rho = −0.427, *P* value < $2.2 \times 10^{-16}$, Supplementary Fig. 13c). For example, the top driver gene for the anterior trajectory *Shox2* was negatively correlated with the posterior trajectory (correlation = −0.333, adjusted *P* value < $2.2 \times 10^{-16}$). On the other hand, *Meox2*, a top driver gene for the posterior trajectory, was negatively correlated with the anterior trajectory (correlation = −0.288, adjusted *P* value < $2.2 \times 10^{-16}$). The osteoblast and dental mesenchymal trajectories shared a list of driver genes, exhibiting high fate probabilities of both trajectories, such as *Runx2* and *Zfpm2* (Supplementary Fig. 13d). Collectively, these data validated the inferred trajectories and driver genes.

## In silico perturbation analysis finds important regulators

Next, we applied CellOracle to assess the impact of perturbing specific regulators on the development of the secondary palate[32]. This algorithm leverages single-cell multiomics data to deduce gene-regulatory networks. It then conducts in silico perturbations to simulate how these changes affect cellular development, relying solely on unperturbed data. Our analysis focused on the cells within the anterior and posterior trajectories. Independent data analysis was conducted, including normalization, clustering, and dimension reduction using PAGA[33] and force-directed graphs, followed by diffusion pseudotime calculation. The manifold revealed two distinct trajectories originating from the multipotent cells towards the anterior and posterior cells (Fig. 6a). We then calculated perturbation scores for all detected TFs. High perturbation scores indicate that in silico knockout of the TF significantly decreased the development of the trajectory, suggesting that the TF is an essential regulator of the trajectory. Interestingly, while the CellOracle perturbation scores were correlated for many TFs, SHOX2 and MEOX2 showed relatively high specificity for the anterior and posterior trajectories, respectively (Fig. 6b). The regulatory networks predicted by CellOracle for all TFs can be found in Supplementary Data 1. Indeed, in silico perturbation of *Shox2* and *Meox2* reversed the developmental velocities for the anterior and posterior trajectories, respectively (Fig. 6c, d and Supplementary Fig. 14).

To further confirm the relevance of Shox2 and Meox2 in secondary palate development, we applied an additional computational algorithm, SCENIC+[34], to infer major regulators of the developmental trajectory (Supplementary Fig. 15). This analysis identified *Meox2* as the top regulon for the posterior trajectory. Due to limited annotation, SCENIC+ does not contain a Shox2 regulon. However, the top regulon for the anterior trajectory was Nfia, which targets *Shox2* based on the Scenicplus annotation. These results further confirmed the importance of both Shox2 and Meox2 in the anterior and posterior trajectories, respectively.

For SHOX2 and MEOX2, CellOracle predicted 11 and 4 target genes, respectively (*P* value < 0.001). It is noteworthy that among these predicted targets, *Satb2*[35,36], *Prrx1*[37,38], and *Prickle1*[39,40] have previously

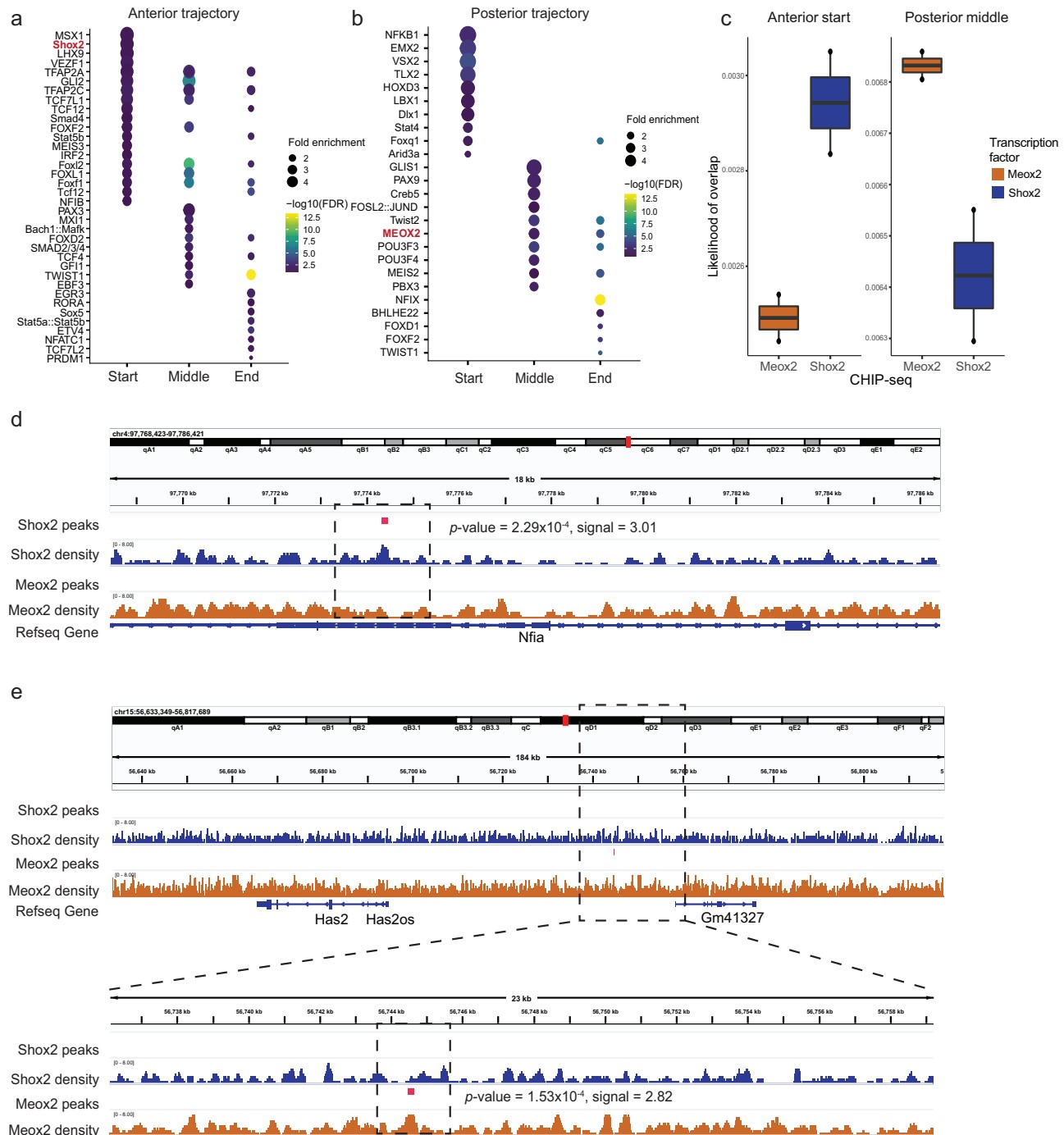

**Fig. 5 | Key transcription factors driving cells towards anterior and posterior states are validated in ChIP-seq experiments. a, b** Dot plots show enriched motifs (*y* axis) at different stages of the **a** anterior trajectory (*x* axis) and **b** posterior trajectories. Dots are scaled by motif enrichment ratio and colored by significance. **c** Left, boxplot shows the likelihood of peaks mapping near genes upregulated at the start of the anterior trajectory (*y* axis) for SHOX2 and MEOX2 binding (*x* axis). Right, boxplot shows the likelihood of peaks mapping near genes upregulated at the middle of the posterior trajectory (*y* axis) for SHOX2 and MEOX2 binding (*x* axis). *n* = 52,797 peaks called from two independent experiments. Minima: Lower limit of the whisker. Maxima: Upper limit of the whisker. Centre: Median line inside the box. The upper and lower box bounds represent the 25% and 75% percentile of data. **d** Integrative Genomics Viewer (IGV) view depicts ChIP-seq binding peak for SHOX2 in the predicted SHOX2 target *Nfia*. **e** IGV view presents ChIP-seq binding peak for MEOX2 upstream of the predicted MEOX2 target *Has2*. The "bdg" (bedGraph) statistical model in the MACS2 tool is utilized, modeling the read counts in the ChIP-seq and control samples as two independent Poisson distributions. Significance of enrichment is determined through the likelihood ratio between these two Poisson distributions. Multiple test adjustments are performed, and the adjusted *P* values are labeled on the plot.

been linked to CP (Fig. 6e). We expanded the MEOX2 network, by utilizing a less stringent *P* value threshold (*P* value <0.01) and identified a total of 39 target genes. Next, we integrated this list of genes with the CleftGeneDB database[5] which collects curated genes with experimental evidence for relevance in cleft palate. Importantly, 6 of these 39 MEOX2 targets have previously been associated with cleft palate, which represents a significantly larger overlap than expected by chance (Fisher's Exact test, *P* value = 0.0002, odds ratio = 9.2,

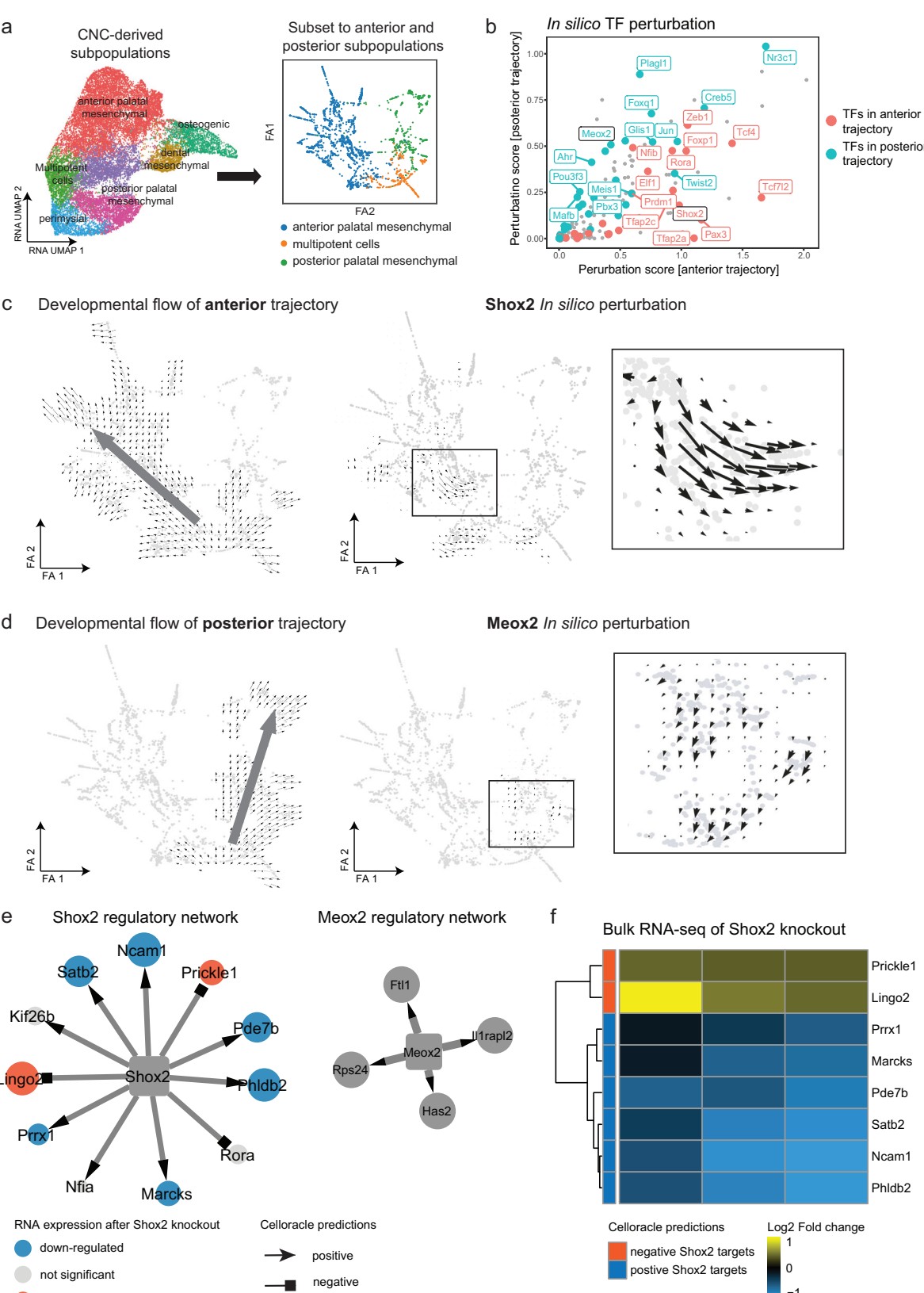

**a** CNC-derived subpopulations — Subset to anterior and posterior subpopulations

- anterior palatal mesenchymal
- multipotent cells
- posterior palatal mesenchymal

**b** *In silico* TF perturbation

- TFs in anterior trajectory
- TFs in posterior trajectory

**c** Developmental flow of **anterior** trajectory — **Shox2** *In silico* perturbation

**d** Developmental flow of **posterior** trajectory — **Meox2** *In silico* perturbation

**e** Shox2 regulatory network — Meox2 regulatory network

RNA expression after Shox2 knockout
- down-regulated
- not significant
- up-regulated

Celloracle predictions
- positive
- negative

**f** Bulk RNA-seq of Shox2 knockout

Celloracle predictions
- negative Shox2 targets
- postive Shox2 targets

Log2 Fold change

Supplementary Fig. 16a). For instance, *Cacna1d* has been reported to be related to orofacial cleft in the GWASdb SNP-Phenotype Associations dataset[41]. In addition, *Foxp2* is linked to nonsyndromic cleft lip and/or palate through genome-wide linkage analysis[42]. Notably, our MEOX2 CHIP-seq data provided further evidence of MEOX2 binding to promoter regions of these genes (Supplementary Fig. 16b).

To validate the predicted gene-regulatory dynamics, we re-analyzed published bulk RNA-seq data derived from the E14.5 anterior hard palatal tissues of *Shox2*[Cre/−] mice, wherein the *Shox2* gene had been knocked out[31]. Among the 11 predicted SHOX2 targets, 8 genes exhibited significantly altered expression following *Shox2* knockout (adjusted *P* value < 0.05) (Fig. 6f). As expected, genes predicted to be

**Fig. 6 | In silico perturbation analysis reveals SHOX2 and MEOX2 as important regulators of the anterior and posterior trajectories, respectively. a** Left: UMAP visualization of CNC-derived mesenchymal subpopulations. Right: Force-directed graph visualizations of anterior and posterior subpopulations. **b** Scatter plot shows in silico TF perturbation scores for the anterior (*x* axis) and posterior (*y* axis) trajectories, respectively. **c** Left: CellOracle vector field graphic shows the developmental flow of anterior trajectory. Arrows start from multipotent cells and point towards the anterior subpopulation. Right: CellOracle vector field graphic shows simulated vector shift after in silico *Shox2* knockout. The developmental flow

towards anterior cells is reversed upon in silico knockout of *Shox2*. **d** Same visualization as in panel C, showing simulated Meox2 perturbation in the posterior trajectory. **e** Network graphs visualized using Cytoscape show predicted SHOX2 and MEOX2 regulatory networks. The shape of the arrowheads is based on the predicted direction of regulation. Target nodes are colored based on regulation in *Shox2*-knockout RNA-seq data. **f** Heatmap shows log2 fold change of RNA expression in three *Shox2* knockout samples normalized to controls. The bulk RNA-seq data used here was downloaded from ref. [31].

positively regulated by *Shox2* demonstrated decreased expression upon *Shox2* knockout (Fig. [6]e, f). Correspondingly, genes predicted to be negatively regulated displayed increased expression upon *Shox2* knockout (Fig. [6]e, f). Taken together, these results suggest that SHOX2 and MEOX2 serve as crucial regulators driving the development of the anterior and posterior secondary palate, respectively.

## Discussion

Dynamic gene expression patterns, driven by the dynamic activity of TFs and accessibility of their binding sites, underlie the formation of the secondary palate. In this study, we generated time-series single-cell multiomics datasets of the mouse secondary palate from E12.5 to E14.5 to dissect lineage-determining TFs that govern the developmental process. Our study profiled multiple modalities of developing mouse secondary palate within the same cells at single-cell resolution.

Cell-type annotation is challenging due to the rareness of data for secondary palate development. The lack of reference data makes automated cell type annotation approaches on scRNA-seq data, such as SingleR[43] and deCS[44], not applicable. In this study, we defined cell types through extensive manual curation of marker genes, which were validated using an independent scRNA-seq dataset taken from a similar tissue[10]. We discovered subpopulations in CNC-derived mesenchymal cells that align with the in vivo anatomical locations, which were validated using in situ hybridization, quantitative RT-PCR, and bulk RNA-seq. The established cell types and subtype-specific gene expression profiles, together with the chromatin accessibility profiles, can be used as a reference to facilitate cell-type annotations in future analyses.

Trajectory analysis in time-series scRNA-seq datasets requires integration of pseudotime with time-point information, where traditional pseudotime approaches are not applicable[45]. In this study, we applied the Wadding-Optimal Transport algorithm[27] to connect cells between adjacent time points and reconstructed five trajectories, demonstrating the continuous developmental landscape of cell states. We identified several driver genes which were previously linked to craniofacial biology. *Msx1*, a driver gene for dental mesenchymal trajectory, was shown to regulate the cell proliferation of dental mesenchymal cells[46] and tooth morphogenesis[47]. Runt-related transcription factor 2 (*Runx2*) is known to regulate tooth and bone formation during the differentiation of CNC-derived cells[48], which exhibited high fate correlations to both dental and osteoblast trajectories in our study. Another key regulator during the middle of the anterior palatal trajectory, *Runx1*, was reported to regulate the fusion and its deficiency caused cleft in the anterior palate[49]. *Dlx1* and *Dlx2* were identified as the top regulators for the posterior palatal mesenchymal trajectory. Concordant with our findings, *Dlx1/2* double knockout mice developed CP due to the vertical growth failure of posterior palatal shelves[50].

Inference of gene-regulatory networks followed by in silico perturbation revealed several regulators of the anterior and posterior trajectories. We focused our analysis on the predicted regulators SHOX2 and MEOX2 and experimentally validated our predictions. *Meox2* was predicted to be a driver gene for the posterior palatal trajectory and *Meox2* null and heterozygous knockout mice exhibited posterior CP due to post-fusion breakdown of palatal shelves[51]. Consistent with our results, *Shox2* null mice exhibited CP that were

confined to the anterior region[52]. Importantly, the posterior palate in *Shox2* null mice was intact, underscoring the gene expression and regulation differences along the anterior-posterior axis of the palate. Our analysis solidifies the role of *Shox2* in establishing the anterior-posterior polarity of the palatal shelves and provides insight into the underlying mechanisms at single-cell resolution.

While our analysis focused on these two important regulators of secondary palate development, our results provide additional information at genome-scale using a single-cell multiome approach. In addition to Shox2 and Meox2, we provide inferred regulatory networks for several additional transcription factors, which represent a valuable resource to the research community (Supplementary Data 1).

Although our study revealed dynamic gene regulation programs, it has limitations in explaining the three-dimensional processes, such as the reorientation of palatal shelves. Therefore, it would be interesting to integrate our results with spatial information, such as 10x Genomics Visium technology that overlay gene expression data with the morphological context in tissues[53], to reveal spatial expression patterns and elucidate the mechanisms of palatal elevation and reorientation at the molecular level. Single-cell proteome data can also be integrated to quantify the downstream protein levels during development[54]. Furthermore, in vivo validations in developing mouse embryos are needed to confirm the regulatory role of identified lineage-determining TFs, such as the knockout of specific TFs or lineage tracing experiments.

In conclusion, our single-cell multiomics atlas of the developing mouse secondary palate charted epigenetic and transcriptional dynamics during palatogenesis and provides a unique resource for the community to facilitate future research of CP.

## Methods

### Tissue preparation, dissociation, and nuclei extraction

All animal procedures and study protocols were approved by the Animal Welfare Committee (AWC) and the Institutional Animal Care and Use Committee (IACUC) of UTHealth (AWC 22-0087). Palatal shelves were isolated from time-mated C57BL/6J mice (000664, Jackson Laboratory) at four distinct developmental timepoints, including embryonic day (E)12.5, E13.5, E14.0, and E14.5. Specifically, at E12.5, we primarily captured the early stages of palatal shelves, during which they emerge as outgrowths from the maxillary processes. E13.5 corresponds to a phase characterized by palatal shelf downgrowth towards the tongue. At E14.0, our samples included palatal shelves in the process of elevation. At this stage, they undergo a significant transformation, transitioning into a more horizontally oriented position above the tongue. Finally, at E14.5, we isolated palatal shelves during the fusion stage, representing the point at which these structures come into contact and ultimately merge along the midline. All mice were maintained in the animal facility of UTHealth under a 12-h light/dark cycle and access to food/water ad libitum.

Single-cell suspensions were prepared from pooled paired secondary palatal shelves of three embryos at E12.5, two embryos at E13.5, and one embryo at E14.0 and E14.5, respectively. The microdissected palatal shelves were treated with 0.25% trypsin and 0.05% EDTA (150 μL) for 5 min at 37 °C with gentle agitation (300 rpm). The dissociated cell mixtures were then suspended with 300 μL Dulbecco's

Modified Eagle's Medium (DMEM, Millipore Sigma) supplemented with heat-inactivated 10% fetal bovine serum (FBS). The cells were centrifuged at 500 g for 5 min at 4 °C and the cell pellets were resuspended and incubated with chilled 300 μL of 0.1× lysis buffer [10 mM Tris-HCl pH 7.4, 10 mM NaCl, 3 mM MgCl₂, 1% BSA, 1 mM DTT, 1 U/μL RNase inhibitor, 0.01% Tween-20, 0.01% Nonidet P40 Substitute, and 0.001% Digitonin] for 3 min on ice, which was stopped with chilled 300 μL wash buffer [10 mM Tris-HCl pH 7.4, 10 mM NaCl, 3 mM MgCl₂, 1% BSA, 1 mM DTT, 1 U/μL RNase inhibitor, and 0.1% Tween-20]. The cells were then collected by centrifugation at $500 \times g$ for 5 min at 4 °C, rinsed with 200 μL wash buffer twice, and re-suspended in Diluted Nuclei Buffer [1× Nuclei Resuspension Buffer, 1 mM DTT, and 1 U/μL RNase inhibitor]. Isolated single-cell nuclei were filtered using a cell strainer (40-μm pore size) and inspected under a microscope to ensure they were successfully dissociated into single cells for subsequent sequencing.

### Single-cell multiome data generation

The single-cell libraries were constructed by following the 10x Genomics Chromium Next GEM Single Cell Multiome ATAC + Gene Expression protocol (CG000338). Briefly, nuclei suspensions were incubated with a transposase, which fragmented the DNA in open regions of the chromatin and added the adapter sequences to the ends of the DNA fragments. The transposed nuclei were loaded onto Chromium Next GEM Chip J (PN-1000234, 10x Genomics, Pleasanton, CA) with partitioning oil and barcoded single-cell gel beads, followed by PCR amplification. The ATAC library and the gene expression library were then prepared separately. The quality of the libraries was examined using the Agilent High Sensitive DNA Kit (#5067-4626) by Agilent Bioanalyzer 2100 (Agilent Technologies, Santa Clara, USA). The library concentrations were determined by qPCR using Collibri Library Quantification kit (#A38524500, ThermoFisher Scientific) on a QuantStudio3 (ThermoFisher Scientific). We then pooled the libraries evenly and performed the paired-end sequencing on an Illumina NextSeq 550 System (Illumina, Inc., USA) using the High Output Kit v2.5 (#20024907, Illumina, Inc., USA).

### Single-cell multiome data processing

The 10x Genomics Cell Ranger ARC (v2.0.0) pipeline was used to process the multiome data. Raw sequencing data were first converted to fastq format using "cellranger-arc mkfastq". The raw files of RNA-seq and ATAC-seq libraries from the same sample were aligned to the UCSC mouse genome (mm10) and quantified using "cellranger-arc count". Samples were aggregated using "cellranger-arc aggr" to normalize the sequencing depth.

The raw RNA count matrix and ATAC fragment data were further processed using R packages Seurat (v4.0.3)[55] and Signac (v1.5.0)[56], respectively. Filtering based on RNA-assay metrics (200<nCount_RNA < 100,000, nFeature_RNA < 7500, percent.mt <20) and ATAC-assay metrics (200 <nCount_ATAC < 100,000, nucleosome_signal <2, TSS.enrichment > 1) resulted in 37,329 cells. The average depth is 73,521 reads per cell, yielding an average of 2472 genes per cell. The gene expression count matrix was then normalized using SCTransform. Principal component (PC) analysis was based on the top 3000 highly variable features. Uniform Manifold Approximation and Projection (UMAP) visualization was constructed using the first 30 PCs.

For the ATAC data, peak calling was performed using MACS2 package with *CallPeaks* function in Signac. Peaks that overlapped with genomic blacklist regions for the mm10 genome were removed[57]. Each peak represents one potential regulatory DNA element. The peak count matrix was then normalized using Latent Semantic Indexing (LSI), including term-frequency (TF) inverse-document frequency (IDF), and Singular value decomposition (SVD). The first LSI component is removed from the downstream analysis as it was highly correlated with sequencing depth. The gene activity was quantified using *GeneActivity* function in Signac (version1.5.0), which aggregated chromatin accessibility intersecting the gene body and promoter regions.

### Projection of external dataset onto our scRNA-seq manifold

To validate the annotated major cell types, we downloaded a scRNA-seq dataset of mouse soft palate, which is the posterior third of the palate, from a recent publication[10]. SCTransform normalization was conducted, followed by PC analysis. The first 30 PCs were used to find anchors between these two datasets using *FindTransferAnchors* function in the Seurat package. The *RunUMAP* function was used to calculate the UMAP coordinates of our dataset with parameters stored in the object (return.model = TRUE). The *MapQuery* function was then used to calculate the coordinates of the external dataset using the same 'uwot' model parameters.

### In situ hybridization

The E14.5 C57BL/6J mouse embryos ($n = 6$) were dissected from a time-pregnant mother and fixed in 4% paraformaldehyde overnight at 4 °C, dehydrated in a graded ethanol series, and embedded in paraffin. Paraffin sections were cut at 4 μm thickness under RNase-free conditions. In situ hybridization was performed using the RNAscope 2.5 Assay platform (ACD, 322360) using specific probes for *Cyp26b1* (ACD, 454241), *Efnb2* (ACD, 477671), *Inhba* (ACD, 455871), *Meox2* (ACD, 823191), *Nrp1* (ACD, 471621), *Prickle1* (ACD, 832641), *Satb2* (ACD, 413261), *Shox2* (ACD, 579051), *Sim2* (ACD, 1110401), and *Trps1* (ACD, 879001). The color images were obtained under a light microscope (BX43, Olympus).

### Quantitative RT-PCR

The anterior ($n = 6$) and posterior ($n = 6$) 1/3 palatal shelves were microdissected from E14.0 C57BL/6J mouse embryos for qRT-PCR. Total RNAs isolated from each region were collected using the QIAshredder and RNeasy mini extraction kit (QIAGEN)[58]. *Gapdh* was used as an internal housekeeping gene control. The PCR primers used in this study are listed in Supplementary Table 2.

### Bulk RNA-seq analysis

The anterior ($n = 3$) and posterior ($n = 3$) 1/3 palatal shelves of E14.0 C57BL/6J mice were microdissected, isolated, and then subjected to bulk RNA-seq. The raw sequenced files were mapped to the mouse reference genome mm10 using HISAT2[59]. StringTie[60] was used to quantify the counts. We then used R package DESeq2 (version 1.30.1) to perform the differential gene expression tests[61]. To project the bulk RNA-seq data into the scRNA principal component space, count matrices from both datasets were first integrated and normalized using *voom* function in the R package limma (version 3.46.0). We performed PCA independently using normalized scRNA data. The normalized bulk RNA-seq data were then projected into the scRNA space using identical principal component gene loadings.

### Peak-gene linkage analysis

We identified peak-gene links using *LinkPeaks* function in Signac[56] based on the approach originally described by SHARE-seq[62]. The Pearson correlation coefficient was calculated between gene expression and peak accessibility. Only peaks within a certain distance (bp) from the gene TSS were included in the model (default: $5 \times 10^5$). The GC content, overall accessibility, and peak size were included in the model as covariates to correct the bias. Benjamini–Hochberg method was used to adjust $P$ values[63]. Only high-confidence peak-gene links with adjusted $P$ value < 0.05 and coefficients >0 were retained for downstream analysis.

## DNA sequence motif enrichment analysis

A total of 746 position weight matrices were loaded from the JASPAR 2020 database[64] using *getMatrixSet* function in TFBSTools package (version 1.32.0, collection = "CORE", tax_group = 'vertebrates'). For a set of differentially accessible peaks, we applied *FindMotifs* function with default parameters to find enriched motifs. Meanwhile, to facilitate the visualization of motif activity, we calculated the motif activity matrix using ChromVAR (version 1.16.0)[65].

## WOT-based terminal state likelihood analysis

The Wadding-Optimal Transport (WOT) was employed to reconstruct the trajectories[27]. Specifically, for a cell at time $t_i$, WOT traced its most likely ancestors and descendants to recover the trajectories by calculating the transition probabilities to cells at time $t_{i+1}$ and $t_{i-1}$. We imported *WOTKernel* from *cellrank.external.kernels* for the following analysis[28]. The growth rates were estimated using the predefined gene proliferation set. The cell–cell transition matrix between adjacent time-points was then calculated using *compute_transition_matrix* function with default parameters (growth_iters=3, growth_rate_key = " growth_rate_init", last_time_point = "connectivities"). The random walks were simulated (n_sims=300). To compute the macrostates, a Generalized Perron Cluster Cluster Analysis (GPCCA) estimator was initialized with WOT connectivity kernel[66]. The inferred macrostates were set as terminal states of five trajectories. The fate probabilities to each terminal state were computed per cell using *compute_absorption_probabilities* function with default parameters (solver = "gmres"). To identify driver genes, we computed the correlation between the fate probabilities and gene expression for each trajectory using *compute_lineage_drivers* function. Multiple testing correction was controlled by the Benjamini–Hochberg method[63]. Only the genes with adjusted $P$ values less than 0.05 and correlations greater than 0.05 were considered driver genes.

## Diffusion pseudotime estimation

We used the built-in function of the Python package Scanpy (version 1.9.1) to estimate the diffusion pseudotime[67]. Specifically, the raw count matrix was loaded in as an AnnData object, followed by standard preprocessing. The neighborhood graph was calculated using *sc.pp.neighbors* function with default parameters (random_state=0). We specified a random cell in CNC-derived multipotent cells (cluster 5) as the root cell. The diffusion pseudotime was then estimated using *scanpy.tl.dpt* function.

## RNA velocity and CellRank analysis

The possorted bam files from Cellranger output were passed to velocyto (version: 0.17.15)[30] to estimate the RNA velocities of single cells. The generated loom file contained data matrices of spliced and unspliced reads and was further processed by scVelo (version 0.2.4)[68]. Seurat-processed gene expression count matrix and UMAP coordinates were converted to "AnnData" object and merged with the velocyto-derived object using *scVelo.utils.merge* function. The merged dataset was filtered using the *scVelo.pp.filter_and_normalize* with default parameters (min_shared_counts = 10, n_top_genes = 2000) and the moments were computed using *scVelo.pp.moments*. The velocity was then calculated using *scVelo.tl.velocity* (mode=stochastic). The estimated velocity vectors were projected and visualized in previously calculated embedding. The initial and terminal state likelihood based on RNA velocity information was estimated using *cellrank.tl.terminal_states* and *cellrank.tl.initial_states* functions with default parameters (weight_connectivities=0.3).

## Trajectory analysis

To identify how and when driver genes were expressed and regulated along each trajectory, we extracted cells with high fate probabilities (fate probability >75% quantile). The extracted cells were then ordered

by diffusion pseudotime. The driver genes were cut into three groups based on quantiles. For each group of driver genes, we conducted gene set enrichment analysis using the R package WebGestalt (version 0.4.4)[69]. The non-redundant Gene Ontology (GO) Biological Process terms were used for pathway annotations. The minimum number of genes in the pathways was set to 5 and the maximum was set to 500. The Benjamini–Hochberg method was used to adjust $P$ values[63]. Those pathways with adjusted $P$ values less than 0.05 were considered statistically enriched. We also performed peak-gene linkage analysis and motif enrichment analysis as described above.

## In silico perturbation analysis

We applied CellOracle (version 0.12.1) to conduct the in silico perturbation analysis following the tutorial provided by the original authors (https://morris-lab.github.io/CellOracle.documentation/)[32]. As recommended, the scRNA-seq data was subsampled to 3000 cells and 3000 highly variable genes for the analysis. Scanpy (version 1.9.3) was used for re-normalization and clustering. Diffusion maps were computed, followed by construction of PAGA and force-directed graphs with default parameters. scATAC-seq data were processed with Cicero (version 1.16.2) with window = 500,000. We retained only those peaks with substantial coaccessibility scores (coaccess ≥ 0.8) for downstream analysis. After inferring the links, a filtering step was implemented using the following parameters: a significance threshold of $P$ = 0.001, "coef_abs" as the weight criterion, and a threshold of 2000 for the number of allowable links. To simulate the effects of TF perturbations, we defined the expression of the perturbed TF as 0. Transition probabilities and embedding shifts were calculated with default parameters (n_neighbors=200, knn_random=True, sigma_corr=0.05). The developmental flow was constructed using inferred diffusion pseudotime. The perturbation effect was then quantified using the inner product of the developmental flow and the perturbation vector. Predicted gene regulatory networks were visualized with Cytoscape (version 3.10.1).

## Re-analysis of published Shox2 knockout bulk RNA-seq data

We downloaded raw counts from the GEO database (accession number GSE129538). The R package DESeq2 (version 1.30.1) was used to identify differentially expressed genes between *Shox2* knockout and wild type samples. Subsequently, the log2 fold change and adjusted $P$ values of predicted SHOX2 and MEOX2 targets were extracted.

## ChIP-seq data analysis

The anterior ($n$ = 2) and posterior ($n$ = 2) 1/3 palatal shelves of E14.0 C57BL/6J mice were microdissected, isolated, and then subjected to ChIP-seq with either SHOX2 antibody (JK-6E) (sc-81955, Santa Cruz), MEOX2 Antibody (A-8) (Sc-376748, Santa Cruz), or normal rabbit IgG (2729, Cell Signaling Technology) as a negative control. The ChIP assays were performed according to the manufacturer's instructions using SimpleChIP® Enzymatic Chromatin IP Kit (Magnetic Beads) (9003, CST). Two independent ChIPs were conducted for library generation for each group. The raw FASTQ datasets were pre-processed using FastQC (version 0.12.1) and mapped to the mouse genome (mm10) using the Bowtie2 alignment algorithm (version 2.5.1). The resulting alignment files were then used as input for MACS2 (version 2.2.9.1) to call peaks with a $P$ value threshold of 0.01. Subsequently, the generated bed files were then converted to bigwig files and visualized using Integrative Genomics Viewer (IGV). Likelihood of overlap was quantified by calculating the rate of peaks falling within 5 kb of the transcription start site of genes up-regulated at the start of anterior and middle of posterior trajectories.

## Re-analysis of published H3K27 acetylation ChIP-seq data

The raw H3K27 acetylation datasets were downloaded from Sequence Read Archive (SRA), comprising 2 samples from anterior palate and

2 samples from posterior palate (SRX6976329)[31]. The tool fastq-dump was used to transfer data to FASTQ format. Subsequently, the raw FASTQ datasets underwent preprocessing through the same pipeline utilized in ChIP-seq data analysis above. To quantify the overlap between scATAC anterior and posterior peaks with the corresponding anterior and posterior acetylation tracks, the findOverlaps function in the GenomicRanges R package (version 1.50.2) was employed. The significance of the overlap was assessed using Fisher's Exact test.

## Statistics and reproducibility
All experiments were performed using with at least $n = 2$ biological replicates for each group. Statistical significance and strength of enrichments were determined using the Wilcoxon rank-sum test. Cells with low quality were excluded from analyses. No statistical method was used to predetermine the sample size.

## Reporting summary
Further information on research design is available in the Nature Portfolio Reporting Summary linked to this article.

## Data availability
The raw and processed single-cell multiome data generated in this study have been deposited in the Gene Expression Omnibus database (GEO) under accession code GSE218576. The bulk RNA-seq data generated in this study are deposited in the GEO under accession code GSE252592. The ChIP-seq data generated in this study are deposited in the GEO under accession code GSE250247. The single-cell RNA-seq data from soft palate is available at the GEO under accession code GSE155928. The raw H3K27 acetylation ChIP-seq data is available at the Sequence Read Archive (SRA) under accession code SRX6976329. Source data are provided with this paper.

## Code availability
All R and Python scripts supporting the findings of this paper are available on the GitHub repository (https://github.com/fangfang0906/Single_cell_multiome_palate).

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

## Acknowledgements

This study was supported by grants from the National Institutes of Health (R01DE030122, R01DE029818 and R01LM012806). F.Y. was a CPRIT Predoctoral Fellow in the Biomedical Informatics, Genomics, and Translational Cancer Research Training Program (BIG-TCR) funded by the Cancer Prevention & Research Institute of Texas (CPRIT RP210045). The sequencing data were generated by the UTHealth Cancer Genomics Core funded by CPRIT (RP180734). The funders had no role in the study design, data collection, analysis, decision to publish, or preparation of the manuscript.

## Author contributions

Conceptualization: Z.Z., J.I. and L.M.S.; methodology: L.M.S., F.Y., C.I., J.I. and Z.Z.; formal analysis: F.Y., L.M.S., A.S., C.I. and G.P.; resources: J.I., A.S., C.I. and Z.Z.; data generation: J.I., C.I., H.Y., X.C., and M.Y.; writing—original draft: F.Y., A.S., C.I., and X.C.; writing—review and editing: Z.Z., L.M.S., J.I., A.S., and C.I.; visualization: F.Y., L.M.S., and CI.; funding acquisition: Z.Z. and J.I.; supervision: Z.Z., J.I., and L.M.S. All authors read and approved the final manuscript.

## Competing interests

The authors declare no competing interests.
