## [Peer Review File · Nature Communications]

REVIEWER COMMENTS

Reviewer #1 (Remarks to the Author):

In their manuscript "Single-cell multiomics decodes regulatory programs for mouse secondary palate development", Yan et al. generated the atlas of developing secondary palate at single cell resolution. For this, the authors relied on single cell multiomics approach, with RNA seq and ATAC seq parts. Finally, the authors suggested that transcription factors *Shox1* and *Dlx1/2* are lineage determining factors for anterior and posterior palatal mesenchymal trajectories. The last statement is not validated by any functional data.

Although this resource will be useful to the community, I feel disappointed with the technical implementation of single cell data analysis, which yielded only basic descriptive information. No regulon analysis or GRN causality inference was performed. Also, I feel frustrated that the authors did not perform any functional validation of their predictions.

Overall, the manuscript is highly descriptive, and, given the routine and easy use of 10X multiome kit, it does not even represent any technical advancement. I do not see what is the discovery here. Publishing such a resource could be ok 3-5 years ago in Nat Comm, but right now it look like a relatively low level. The authors simply invested some funds into generating the data, and then performed quite standard analysis to showcase cell populations, trajectories and factors associated with transitions (using both RNA and ATAC-seq parts).

Despite all these downsides, I do not recommend the rejection at this point and wish to give the authors the second chance, and for this, the authors have to do a lot of work, both experimental and analytical. Below I provide a more precise guidance:

1. Formal trajectories must be generated, and the causality must be inferred via SCENIC+, SCENIC and CellOracle analysis. The GRNs must be shown in dynamics as evolving graphs during pseudotime.
2. The authors must analyse cell fate decisions within the progenitor populations and infer biased states.
3. Most importantly, the authors must use some approach to validate the most key findings in a functional experiment. For this, the authors shall use lentiviral microinjections, CAS9/CRISPR knockouts and analysis in F0 or F1 and so on, and also the regulatory regions reporter assays.

Without extensively addressed point 3, this paper will likely fail to be published in Nature Communications.

Reviewer #2 (Remarks to the Author):

This manuscript details results obtained from combined single-cell RNA-seq and ATAC-seq of mouse secondary palatal shelves from E12.5-E14.5. The authors mostly focus on the cranial neural crest-derived mesenchyme, demonstrating differentiation into anterior palatal mesenchyme, posterior palatal mesenchyme, osteogenic, dental mesenchyme and perimysial lineages, consistent with previous

findings. They further analyze the above data to predict transcription factors that determine individual lineages, such as *Shox2* in the anterior palatal mesenchyme, again consistent with previous findings. The data is robust, clearly presented and will likely be a valuable resource for the field. However, it is unclear what new information is gained from this study, especially given the lack of validation studies and the fact that the majority of transcription factors discussed already have demonstrated *in vivo* roles in various secondary palate cell types.

Major comments:

1. Page 4, line 20: Development of the mouse secondary palate occurs from E11.5-E16.5 (see Bush and Jiang, 2012, PMC3243091). Relatedly, the image of secondary palatal shelves fusing at E14.5 in Figure 1A is premature, as this representation more closely matches E15.0. The authors should clearly discuss why they chose the E12.5, E13.5, E14.0 and E14.5 timepoints and which steps of secondary palate development they were capturing at each timepoint.
2. On page 5, the authors outline the marker gene expression that defined each of the eight cell types, however the majority of these markers are not in Figures 1D and 1E. The authors should include this data and/or explain this discrepancy.
3. It is unclear how the UMAPs in Figure 3B define medial and lateral positions in the cranial neural crest-derived mesenchyme. The authors should address whether the marker genes indicated here (*Osr2*, *Fgf10*, *Fgf7* and *Dlx5*) have previously been shown to have restricted expression in these domains and/or perform *in situ* RNA hybridization experiments.
4. The authors state on page 11 that "...we characterized a list of lineage-determining TFs that control the anterior trajectory, such as *Shox2* at the early stage, *Foxl2* at the middle stage, and *Nr2e1* at the late stage of the trajectory", but do not provide any statistics on the latter two factors. Further, given the long list of transcription factors indicated at the right of Figure 4H, each with relatively low fold enrichment, it is hard to understand how the authors can make such a statement about *Nr2e1*, for example. To definitely back up such statements, the authors would need to perform ChIP-seq for transcription factors of interest and show that each binds at least a subset of the identified motifs, knock down individual transcription factor function *in vivo* and demonstrate that a specific trajectory is affected, and/or perform detailed lineage tracing experiments (for example, using novel Cre lines generated from this data). Without such data, which would significantly enhance the manuscript, the authors need to temper such statements as "...we identified a list of TFs that control each trajectory by binding to regulatory elements to regulate the expression of the above-mentioned driver genes" on page 14.
5. The authors need to better define the novelty of their findings, as the majority of transcription factors discussed already have demonstrated *in vivo* roles in various secondary palate subtypes.

Minor comments:

1. The authors should color-code or otherwise differentiate the data in Figure S1A and B so that readers can assess how well the replicates at each timepoint overlap in each of these metrics. Relatedly, it is difficult to make out the three colors representing the E13.5 replicates in Figure S1E. Further, in Figure

S1D, E13.5 replicates 1 and 2 appear more similar to E14.0 replicate 2 than E13.5 replicate 3. The authors should comment on this discrepancy.

2. Page 4, line 29 and page 5, lines 1-2 refer to Figure S1C top and bottom, when the panels are presented as left and right.

3. The label “Glial” is missing from the y-axis in Figure 1E.

4. The RNA and motif p-values for Twist2 on page 7 do not match the values in Table 1.

5. It is unclear what the top and bottom panels represent at each location in Figure 3D. On page 8, lines 15-16 the authors state that “Inhba was restricted to beneath the epithelial layer in the anterior region”, when a similar expression pattern is also noted in the middle region. Similarly, the authors state that “Sim2 and Trps1 were expressed only in the anterior half of the posterior region”, though they appear to be expressed medially.

6. On page 8, the authors should define how the “anterior” and “posterior” regions of the secondary palate were delineated for bulk RNA-seq.

7. The authors should comment on why the differentially expressed genes with the largest fold changes in bulk RNA-seq in Figure 3F were often not detected in the scRNA-seq. Relatedly, it is unclear to this reviewer what is being represented in Figure 3H. Does this indicate that only ~6 transcripts commonly mapped to anterior and posterior locations between datasets?

8. It is unclear why none of the labeled driver genes for posterior trajectory in Figure 5C appear in Figure 4L, especially Meox2. The colors in the legend of Figure 5C do not appear to match the colors in the graph.

Reviewer #3 (Remarks to the Author):

The manuscript provides a comprehensive analysis of the developmental programs of the mouse secondary palate, utilizing single-cell multiome data to simultaneously profile RNA-seq and ATAC-seq in the same cells. The study identified distinct cell types and developmental trajectories, as well as key regulators and signaling pathways involved in the development of the mouse secondary palate. Overall, I find the paper to be a useful reference and data resource for studying developmental processes at single-cell resolution.

- The cell clustering and annotation appear to be driven solely by signals in RNA data, while recent computational methods (e.g. Seurat v4) can handle paired RNA and ATAC data to create joint embeddings of the two modalities for clustering. Therefore, using information from both modalities for clustering would be more reasonable.

- It would be helpful to understand how the authors selected the 3000 highly variable genes and whether the annotation results are stable to the chosen genes.

- Checking motif enrichment of ATAC data would be beneficial in addition to checking the enrichment of gene activity scores for differentially expressed genes in the annotated cell types.

- How consistent is the diffusion pseudotime against real time (days) in the trajectory analysis?

- The driver genes were inferred by checking marginal correlation with the fate probability, but it should be acknowledged that not all marginally correlated genes are necessarily drivers. There could be many "passenger" genes correlated with the real driver genes, but counted as driver genes using this method. This limitation should be discussed.

Response to Reviewers

MS ID: NCOMMS-23-08175

MS Title: Single-cell multiomics decodes regulatory programs for mouse secondary palate development

Response to Reviewer #1

In their manuscript “Single-cell multiomics decodes regulatory programs for mouse secondary palate development”, Yan et al. generated the atlas of developing secondary palate at single cell resolution. For this, the authors relied on single cell multiomics approach, with RNA seq and ATAC seq parts. Finally, the authors suggested that transcription factors *Shox1* and *Dlx1/2* are lineage determining factors for anterior and posterior palatal mesenchymal trajectories. The last statement is not validated by any functional data.

Although this resource will be useful to the community, I feel disappointed with the technical implementation of single cell data analysis, which yielded only basic descriptive information. No regulon analysis or GRN causality inference was performed. Also, I feel frustrated that the authors did not perform any functional validation of their predictions.

Overall, the manuscript is highly descriptive, and, given the routine and easy use of 10X multiome kit, it does not even represent any technical advancement. I do not see what is the discovery here. Publishing such a resource could be ok 3-5 years ago in Nat Comm, but right now it look like a relatively low level. The authors simply invested some funds into generating the data, and then performed quite standard analysis to showcase cell populations, trajectories and factors associated with transitions (using both RNA and ATAC-seq parts).

Despite all these downsides, I do not recommend the rejection at this point and wish to give the authors the second chance, and for this, the authors have to do a lot of work, both experimental and analytical.

Response: We thank the reviewer #1 for the positive but also critical comments on our work. In our revision, we have addressed your concerns. From an analytical perspective, as suggested by the reviewer, CellOracle was applied to infer gene regulatory networks (GRNs) and predict crucial regulators. From the experimental perspective, we generated and analyzed ChIP-seq data to validate our computational predictions as well as analyzed a published *Shox2*-knockout RNA-seq dataset. In addition, we would like to point out that single-cell multiomics has not yet been applied to the study of palate development. We would also like to note that CellOracle was published in early February 2023, when our original manuscript was submitted.

Below I provide a more precise guidance:

1. Formal trajectories must be generated, and the causality must be inferred via SCENIC+, SCENIC and CellOracle analysis. The GRNs must be shown in dynamics as evolving graphs during pseudotime.

Response: We thank the reviewer for this valuable suggestion. As suggested by the reviewer, we applied CellOracle to our multiome data and performed *in silico* transcription factor (TF) perturbation analysis. This analysis is included in the revised manuscript and the results are summarized in the new Figure 6 and Supplementary Figure 13.

We added the following section to the Results section of the revised manuscript on pages 12 and 13:

***In silico* perturbation analysis analysis finds driver genes**

Next, we applied CellOracle to assess the impact of perturbing specific regulators on the development of the secondary palate. This algorithm leverages single-cell multi-omics data to

deduce gene-regulatory networks. It then conducts *in silico* perturbations to simulate how these changes affect cellular development, relying solely on unperturbed data.

Our analysis focused on the cells within the anterior and posterior trajectories. Independent data analysis was conducted, including normalization, clustering, and dimension reduction using PAGA and force-directed graphs, followed by diffusion pseudotime calculation. The manifold revealed two distinct trajectories originating from the multipotent cells towards the anterior and posterior cells (**Figure 6a**). We then calculated perturbation scores for all detected TFs. High perturbation scores indicate that *in silico* knockout of the TF significantly decreased the development of the trajectory, suggesting that the TF is an essential driver of the trajectory. Interestingly, while the CellOracle perturbation scores were correlated for many TFs, SHOX2 and MEOX2 showed relatively high specificity for the anterior and posterior trajectories, respectively (**Figure 6b**). Indeed, *in silico* perturbation of *Shox2* and *Meox2* reversed the developmental velocities for the anterior and posterior trajectories, respectively (**Figure 6c, d, Supplementary Fig. 13**).

The regulatory networks predicted by CellOracle for all transcription factors can be found in **Supplementary Table 3**. For SHOX2 and MEOX2, CellOracle predicted 11 and 4 target genes, respectively. It is noteworthy that among these predicted targets, *Satb2*, *Prrx1*, and *Prickle1* have previously been linked to cleft palate (**Figure 6e**).”

Figure 6. *In silico* perturbation analysis reveals SHOX2 and MEOX2 as drivers of the anterior and posterior trajectories, respectively. **a** Left: UMAP visualization of CNC-derived mesenchymal subpopulations. Right: Force directed graph visualizations of anterior and posterior subpopulations. **b** Scatter plot shows *in silico* TF perturbation scores for the anterior (x-axis) and posterior (y-axis) trajectories, respectively. **c** Left: CellOracle vector field graphic shows the developmental flow of anterior trajectory. Arrows start from multipotent cells and point towards the anterior subpopulation. Right: CellOracle vector field graphic shows simulated vector shift after *in silico* Shox2 knockout. The developmental flow towards anterior cells is reversed upon *in silico* knockout of Shox2. **d** Same visualization as in panel c, showing simulated Meox2 perturbation in the posterior trajectory. **e** Network graphs visualized using Cytoscape show predicted SHOX2 and MEOX2 regulatory networks. The shape of the arrow heads is based on the predicted direction of regulation. Target nodes are colored based on regulation in *Shox2*-knockout RNA-seq data. **f** Heatmap shows log₂ fold change of RNA expression in three *Shox2* knockout samples normalized to controls. The bulk RNA-seq data used here was downloaded from Xu et al. 2019.

Supplementary Fig. 13. Shox2 and Meox2 exhibit high perturbation score after *in silico* knockout simulation for anterior and posterior trajectories, respectively. **a** Left: UMAP visualization of data subset showing Shox2 expression in anterior subpopulation. Middle: Grid plot showing inner product of Shox2 perturb simulation and anterior trajectory developmental flow. Right: scatter plot shows inner product score (y-axis) along pseudotime (x-axis). **b** Same Visualization as panel A for Meox2 in the posterior trajectory.

2. The authors must analyze cell fate decisions within the progenitor populations and infer biased states.

Response: We thank the reviewer for this valuable suggestion. We would like to point out that in our previous version, the CellRank algorithm used in Supplementary Figure 8 detects the initial and terminal cell states of the system and computes a global map of fate potentials, assigning each cell, including the progenitor populations, the probability of reaching each terminal state.

In addition, as suggested by the reviewer, we compared cell fate decisions within the progenitor populations directly. We added the description of these results to the main text and summarized in a new Supplementary Figure 9. In the revised manuscript, on page 10, we added:

“To more granularly resolve these velocity predictions, we applied CellRank, which infers initial and terminal state probabilities for each cell based on RNA velocity. Consistent with WOT-derived trajectories, CellRank found high initial state probabilities in CNC-derived multipotent cells and high terminal state probabilities in the late-stage subpopulations (**Supplementary Fig. 8**). To analyze cell fate decisions within the progenitor populations, we isolated those with high transition probabilities toward the anterior and posterior trajectories, respectively. Differential gene expression analysis unveiled distinct expression profiles in these two subpopulations. In the progenitor anterior subpopulation, genes such as *Shox2*, *Satb2*, *Inhba*, *Cyp26b1*, and *Nrp1* demonstrated significantly higher expression levels. Conversely, in the progenitor posterior subpopulation, genes like *Meox2*, *Prickle1*, *Sim2*, *Efnb2*, and *Trps1* exhibited elevated expression (**Supplementary Fig. 9**). The results align with the expression profile observed in terminally differentiated anterior and posterior populations, as illustrated in **Figure 3**”.

Supplementary Fig. 9. Progenitor cells with high transition probabilities displayed higher anterior-specific genes expressions. a Scatter plot displays transition probabilities to anterior (x-axis) and posterior (y-axis) states for progenitor cells. **b** Volcano plot shows average log2 fold change (x-axis) and $-\log_{10}$ adjusted p-value (y-axis) of differentially expressed genes between progenitor cells biased towards the anterior and posterior cell fates.

3. Most importantly, the authors must use some approach to validate the most key findings in a functional experiment. For this, the authors shall use lentiviral microinjections, CAS9/CRISPR knockouts and analysis in F0 or F1 and so on, and also the regulatory regions reporter assays. Without extensively addressed point 3, this paper will likely fail to be published in Nature Communications.

Response: We thank the reviewer for this critical point. We agree that functional assays are important. However, we were not able to perform CRISPR/Cas9 knockouts because generation of mouse strains by CRISPR/Cas 9 for the candidate genes/mutations will take many months, but we were given only three months. In addition, this is somewhat out of scope of our current study, which has already been extensive. Please kindly note that, although single-cell multiome technologies are

not very new, there has been no such a study in the field of craniofacial development, a critical development stage important for not only developmental biology but also human diseases. We sincerely hope that you will agree our explanation. Instead, to experimentally validate our key findings, we generated ChIP-seq experiments and included these in our revision.

On pages 11 and 12, we wrote “We next investigated underlying transcriptional regulators by motif enrichment analysis. We characterized a list of lineage-determining TFs that are potentially candidates to control each trajectory by binding to regulatory elements to regulate the expression of the above-mentioned driver genes. *Shox2* was identified as an important regulator at the start of the anterior trajectory (motif adjusted p-value = 6.09×10^{-3} , motif fold change = 4.21, gene adjusted p-value < 2.2×10^{-16} , gene fate correlation = 0.47) (**Figure 5a**). *MEOX2* showed potential regulatory roles in the middle (motif adjusted p-value = 0.024, motif fold change = 2.87) and end of the posterior trajectory (motif adjusted p-value = 1.85×10^{-3} , motif fold change = 2.06, **Figure 5b**). To experimentally validate these predictions, we conducted chromatin immunoprecipitation followed by sequencing (ChIP-seq) experiments. *SHOX2* and *MEOX2* ChIP-seq data were generated from anterior and posterior palate tissue, respectively. We calculated the likelihood of observing a ChIP-seq peak near the genes that were upregulated at the start of the anterior trajectory and the middle of the posterior trajectory. As computationally predicted, the likelihood of observing a *SHOX2* peak was increased for genes up-regulated at the start of the anterior trajectory. Correspondingly, the likelihood of observing a *MEOX2* peak was increased for genes up-regulated at the middle of the posterior trajectory (**Figure 5c**). For example, we observed a ChIP-seq binding peak for *SHOX2* but not *MEOX2* in the predicted *SHOX2* target *Nfia* (**Figure 5d**). Simultaneously, a *MEOX2* binding peak was observed upstream of the predicted *MEOX2* target *Has2* (**Figure 5e**).”

Figure 5. Key transcription factors driving cells towards anterior and posterior states are validated in CHIP-seq experiments. **a, b** Dot plots show enriched motifs (y-axis) at different stages of the **(a)** anterior trajectory (x-axis) and **(b)** posterior trajectories. Dots are scaled by motif enrichment ratio and colored by significance. **c** Left, boxplot shows the likelihood of peaks mapping near genes upregulated at the start of the anterior trajectory (y-axis) for SHOX2 and MEOX2 binding (x-axis). Right, boxplot shows the likelihood of peaks mapping near genes upregulated at the middle of the posterior trajectory (y-axis) for SHOX2 and MEOX2 binding (x-axis). **d** Integrative Genomics Viewer (IGV) view depicts ChIP-seq binding peak for SHOX2 in the predicted SHOX2 target *Nfia*. **e** IGV view presents ChIP-seq binding peak for MEOX2 upstream of the predicted MEOX2 target *Has2*.

In addition, we re-analyzed published RNA-seq data of *Shox2* knockout to validate our *in silico* perturbation predictions in the revision on page 13:

“To validate the predicted gene-regulatory dynamics, we re-analyzed published bulk RNA-seq data derived from the E14.5 anterior hard palatal tissues of *Shox2*^{C^{re}/-} mice, wherein the *Shox2* gene had been knocked out. Among the 11 predicted SHOX2 targets, 8 genes exhibited significantly altered expression following *Shox2* knockout (adjusted p-value < 0.05) (**Figure 6f**). As expected, genes predicted to be positively regulated by *Shox2* demonstrated decreased expression upon *Shox2* knockout (**Figure 6e, f**). Correspondingly, genes predicted to be negatively regulated displayed increased expression upon *Shox2* knockout (**Figure 6e, f**). Taken together these results suggest that SHOX2 and MEOX2 serve as crucial regulators driving the development of the anterior and posterior secondary palate, respectively.”

Figure 6. *In silico* perturbation analysis reveals SHOX2 and MEOX2 as drivers of the anterior and posterior trajectories, respectively. ...f Heatmap shows log₂ fold change of RNA expression for predicted targets in three *Shox2* knockout samples normalized to controls. The bulk RNA-seq data used here was downloaded from Xu et al. 2019.

Together with several experiments included in the previous version (bulk RNA-seq, *in situ* hybridization, quantitative RT-PCR, etc.), we hope that the reviewer will agree that our main findings are validated and that this original work is a novel contribution in craniofacial development field.

Response to Reviewer #2

This manuscript details results obtained from combined single-cell RNA-seq and ATAC-seq of mouse secondary palatal shelves from E12.5-E14.5. The authors mostly focus on the cranial neural crest-derived mesenchyme, demonstrating differentiation into anterior palatal mesenchyme, posterior palatal mesenchyme, osteogenic, dental mesenchyme and perimysial lineages, consistent with previous findings. They further analyze the above data to predict transcription factors that determine individual lineages, such as *Shox2* in the anterior palatal mesenchyme, again consistent with previous findings. The data is robust, clearly presented and will likely be a valuable resource for the field. However, it is unclear what new information is gained from this study, especially given the

lack of validation studies and the fact that the majority of transcription factors discussed already have demonstrated *in vivo* roles in various secondary palate cell types.

Response: We thank the reviewer #2 for carefully reading our manuscript and summarizing our work nicely. We understood some results of our work are consistent with previous findings, which demonstrated that our single-cell multiomics approach is reliable and informative. However, our study investigated craniofacial development at the single-cell multiomics level giving insight into the underlying mechanisms at unprecedented resolution. The extensive data and analysis will provide important knowledge and further understanding of mechanisms underlying craniofacial disease.

Due to the limited scope of the current work, we were not able to develop mouse strains or cell lines for additional experimental validation. Such experiments will take many months, but we were given only three months for revision. Please kindly note that, although single-cell multiome technologies are not very new, there has been no such a study in the field of craniofacial development, a critical development stage important for not only developmental biology but also human diseases. We sincerely hope that you will agree our explanation. Instead, we performed ChIP-seq experiments to validate our key findings (*Shox2* and *Meox2*) and re-analyzed published RNA-seq of *Shox2* knockout in the revision. Together with several experiments included in the previous version (bulk RNA-seq, *in situ* hybridization, quantitative RT-PCR, etc.), we hope that the reviewer will agree that our main findings are validated and that this original work is a novel contribution in craniofacial development field.

Major comments:

1. Page 4, line 20: Development of the mouse secondary palate occurs from E11.5-E16.5 (see Bush and Jiang, 2012, PMC3243091). Relatedly, the image of secondary palatal shelves fusing at E14.5 in Figure 1A is premature, as this representation more closely matches E15.0. The authors should clearly discuss why they chose the E12.5, E13.5, E14.0 and E14.5 timepoints and which steps of secondary palate development they were capturing at each timepoint.

Response: We thank the reviewer for the valuable comment. We selected the specific timepoints based on the mouse anatomy described by the FaceBase project. Color-coded scanning electron microscopy images show the fusing of the palatal shelves at 14.5:

Screenshot taken from FaceBase website

(<https://www.facebase.org/resources/mouse/mouseanatomy/>) shows fusing of palatal shelves at E14.5.

To make it more clear why we selected E12.5, E13.5, E14.0, and E14.5 timepoints, we added the following text to the main text of the revised manuscript. On page 4, we wrote: “To dissect gene regulation mechanisms at the cellular level in the developing mouse secondary palate, we performed single-cell multiome sequencing using the 10x Chromium Single Cell Multiome platform. Following major developmental milestones of the mouse secondary palate as defined by the Facebase consortium, we generated scRNA-seq and scATAC-seq libraries from the same cells at E12.5 (n=2), E13.5 (n=3), E14.0 (n=2), and E14.5 (n=2) (**Figure 1a**).”

2. On page 5, the authors outline the marker gene expression that defined each of the eight cell types, however the majority of these markers are not in Figures 1D and 1E. The authors should include this data and/or explain this discrepancy.

Response: We thank the reviewer for pointing this issue out. The markers in Figure 1 were selected from top differentially expressed genes while we outlined those from publications mentioned in the main text. We have updated Figure 1d and 1e to make them consistent in the revised version. The main text was also updated.

Figure 1. Single-cell multiome assays dissect transcriptome and epigenome changes of the developing mouse secondary palate. ... (d, e) Dot plot illustrates marker gene expression (x-axis) (d) and gene activity (e) (x-axis) across cell types (y-axis). Dot size is proportional to the percent of expressed cells. Colors indicate low (purple) to high (yellow) expression.

On page 5, we wrote “Each cell type was defined by canonical marker gene expressions, including CNC-derived mesenchymal cells (*Prrx1*, $n=28,529$, 78.91%), epithelial cells (*Krt14*, $n=5,866$, 16.23%), endothelial (*Cdh5*, *Cldn5*, $n=714$, 1.97%), myeloid (*Lyz2*, $n=397$, 1.10%), glial cells (*Plp1*, *Sox10*, $n=307$, 0.85%), myogenic precursors (*Myf5*, $n=200$, 0.55%), neuronal (*Tubb3*, *Stmn2*, $n=113$, 0.31%) and myocytes (*Myh7*, $n=28$, 0.08%) (Figure 1d, Supplementary Fig. 3a).”

3. It is unclear how the UMAPs in Figure 3B define medial and lateral positions in the cranial neural crest-derived mesenchyme. The authors should address whether the marker genes indicated here (*Osr2*, *Fgf10*, *Fgf7* and *Dlx5*) have previously been shown to have restricted expression in these domains and/or perform in situ RNA hybridization experiments.

Response: We apologize for not being clear. These markers were previously shown to be expressed in lateral or medial regions. In the manuscript by Lan et al., the authors stated “From E12.5 to E13.5, *Osr2* mRNA is expressed abundantly throughout the palatal mesenchyme, with lateral regions expressing higher levels than the medial regions.” Similarly, in the manuscript by Levi et al., the authors reported “Co-expression of *Msx1* and *Dlx5* is limited to the most lateral region of the maxillary mesenchyme.” However, in the revised manuscript, we focused on anterior and posterior subpopulations, so the results related to medial and lateral positions were removed from the main text.

References

- Lan Y, Ovitt CE, Cho E-S, Maltby KM, Wang Q, Jiang R. Odd-skipped related 2 (*Osr2*) encodes a key intrinsic regulator of secondary palate growth and morphogenesis. *Development* 131, 3207-3216 (2004).
- Levi, G., Mantero, S., Barbieri, O., Cantatore, D., Paleari, L., Beverdam, A., ... & Merlo, G. R. (2006). *Msx1* and *Dlx5* act independently in development of craniofacial skeleton, but converge on the regulation of Bmp signaling in palate formation. *Mechanisms of development*, 123(1), 3-16.

4. The authors state on page 11 that “...we characterized a list of lineage-determining TFs that control the anterior trajectory, such as *Shox2* at the early stage, *Foxl2* at the middle stage, and *Nr2e1* at the late stage of the trajectory”, but do not provide any statistics on the latter two factors. Further, given the long list of transcription factors indicated at the right of Figure 4H, each with relatively low fold enrichment, it is hard to understand how the authors can make such a statement about *Nr2e1*, for example. To definitely back up such statements, the authors would need to perform

ChIP-seq for transcription factors of interest and show that each binds at least a subset of the identified motifs, knock down individual transcription factor function in vivo and demonstrate that a specific trajectory is affected, and/or perform detailed lineage tracing experiments (for example, using novel Cre lines generated from this data). Without such data, which would significantly enhance the manuscript, the authors need to temper such statements as "...we identified a list of TFs that control each trajectory by binding to regulatory elements to regulate the expression of the above-mentioned driver genes" on page 14.

Response: We thank the reviewer for this constructive suggestion. We agree that additional analysis was necessary to specify lineage determining transcription factors. In the revision, we added *in silico* perturbation analysis using CellOracle, a novel computational method to predict the impact of perturbation of transcription factors on cellular development. This analysis is included in the revised manuscript and the results are summarized in the new Figure 6 and Supplementary Figure 12 on pages 12 and 13:

"*In silico* perturbation analysis finds driver genes

Next, we applied CellOracle to assess the impact of perturbing specific regulators on the development of the secondary palate. This algorithm leverages single-cell multi-omics data to deduce gene-regulatory networks. It then conducts *in silico* perturbations to simulate how these changes affect cellular development, relying solely on unperturbed data.

Our analysis focused on the cells within the anterior and posterior trajectories. Independent data analysis was conducted, including normalization, clustering, and dimension reduction using PAGA and force-directed graphs, followed by diffusion pseudotime calculation. The manifold revealed two distinct trajectories originating from the multipotent cells towards the anterior and posterior cells (**Figure 6a**). We then calculated perturbation scores for all detected TFs. High perturbation scores indicate that *in silico* knockout of the TF significantly decreased the development of the trajectory, suggesting that the TF is an essential driver of the trajectory. Interestingly, while the CellOracle perturbation scores were correlated for many TFs, SHOX2 and MEOX2 showed relatively high specificity for the anterior and posterior trajectories, respectively (**Figure 6b**). Indeed, *in silico* perturbation of *Shox2* and *Meox2* reversed the developmental velocities for the anterior and posterior trajectories, respectively (**Figure 6c, d, Supplementary Fig. 13**).

The regulatory networks predicted by CellOracle for all transcription factors can be found in **Supplementary Table 3**. For SHOX2 and MEOX2, CellOracle predicted 11 and 4 target genes, respectively. It is noteworthy that among these predicted targets, *Satb2*, *Prrx1*, and *Prickle1* have previously been linked to cleft palate (**Figure 6e**).

Figure 6. *In silico* perturbation analysis reveals SHOX2 and MEOX2 as drivers of the anterior and posterior trajectories, respectively. **a** Left: UMAP visualization of CNC-derived mesenchymal subpopulations. Right: Force directed graph visualizations of anterior and posterior subpopulations. **b** Scatter plot shows *in silico* TF perturbation scores for the anterior (x-axis) and posterior (y-axis) trajectories, respectively. **c** Left: CellOracle vector field graphic shows the developmental flow of anterior trajectory. Arrows start from multipotent cells and point towards the anterior subpopulation. Right: CellOracle vector field graphic shows simulated vector shift after *in silico* Shox2 knockout. The developmental flow towards anterior cells is reversed upon *in silico* knockout of Shox2. **d** Same visualization as in panel c, showing simulated Meox2 perturbation in the posterior trajectory. **e** Network graphs visualized using Cytoscape show predicted SHOX2 and MEOX2 regulatory networks. The shape of the arrow heads is based on the predicted direction of regulation. Target nodes are colored based on regulation in *Shox2*-knockout RNA-seq data. **f** Heatmap shows log₂ change of RNA expression in three *Shox2* knockout samples normalized to controls. The bulk RNA-seq data used here was downloaded from Xu et al. 2019.

Supplementary Fig. 13. Shox2 and Meox2 exhibit high perturbation score after *in silico* knockout simulation for anterior and posterior trajectories, respectively. **a** Left: UMAP visualization of data subset showing Shox2 expression in anterior subpopulation. Middle: Grid plot showing inner product of Shox2 perturb simulation and anterior trajectory developmental flow. Right: scatter plot shows inner product score (y-axis) along pseudotime (x-axis). **b** Same Visualization as panel A for Meox2 in the posterior trajectory.

In addition, we re-analyzed published RNA-seq data of *Shox2* knockout to validate our *in silico* perturbation predictions in the revision on page 13:

“To validate the predicted gene-regulatory dynamics, we re-analyzed published bulk RNA-seq data derived from the E14.5 anterior hard palatal tissues of *Shox2^{Cre/-}* mice, wherein the *Shox2* gene had been knocked out. Among the 11 predicted SHOX2 targets, 8 genes exhibited significantly altered expression following *Shox2* knockout (adjusted p-value < 0.05) (**Figure 6f**). As expected, genes predicted to be positively regulated by *Shox2* demonstrated decreased expression upon *Shox2* knockout (**Figure 6e, f**). Correspondingly, genes predicted to be negatively regulated displayed increased expression upon *Shox2* knockout (**Figure 6e, f**). Taken together these results suggest that SHOX2 and MEOX2 serve as crucial regulators driving the development of the anterior and posterior secondary palate, respectively.”

Figure 6. *In silico* perturbation analysis reveals SHOX2 and MEOX2 as drivers of the anterior and posterior trajectories, respectively. ...f Heatmap shows log₂ fold change of RNA expression for predicted targets in three *Shox2* knockout samples normalized to controls. The bulk RNA-seq data used here was downloaded from Xu et al. 2019.

Finally, we also performed ChIP-seq experiments, as suggested by the reviewer, to validate our key findings (*Shox2* and *Meox2*). On page 12, we wrote “To experimentally validate these predictions, we conducted chromatin immunoprecipitation followed by sequencing (ChIP-seq) experiments. SHOX2 and MEOX2 ChIP-seq data were generated from anterior and posterior palate tissue, respectively. We calculated the likelihood of observing a ChIP-seq peak near the genes that were upregulated at the start of the anterior trajectory and the middle of the posterior trajectory. As computationally predicted, the likelihood of observing a SHOX2 peak was increased for genes upregulated at the start of the anterior trajectory. Correspondingly, the likelihood of observing a MEOX2 peak was increased for genes up-regulated at the middle of the posterior trajectory (**Figure 5c**). For example, we observed a ChIP-seq binding peak for SHOX2 but not MEOX2 in the predicted SHOX2 target *Nfia* (**Figure 5d**). Simultaneously, a MEOX2 binding peak was observed upstream of the predicted *Meox2* target *Has2* (**Figure 5e**).”

Figure 5. Key transcription factors driving cells towards anterior and posterior states are validated in CHIP-seq experiments. **a, b** Dot plots show enriched motifs (y-axis) at different stages of the **(a)** anterior trajectory (x-axis) and **(b)** posterior trajectories. Dots are scaled by motif enrichment ratio and colored by significance. **c** Left, boxplot shows the likelihood of peaks mapping near genes upregulated at the start of the anterior trajectory (y-axis) for SHOX2 and MEOX2 binding (x-axis). Right, boxplot shows the likelihood of peaks mapping near genes upregulated at the middle of the posterior trajectory (y-axis) for SHOX2 and MEOX2 binding (x-axis). **d** Integrative Genomics Viewer (IGV) view depicts CHIP-seq binding peak for SHOX2 in the predicted SHOX2 target *Nfia*. **e** IGV view presents CHIP-seq binding peak for MEOX2 upstream of the predicted MEOX2 target *Has2*.

5. The authors need to better define the novelty of their findings, as the majority of transcription factors discussed already have demonstrated *in vivo* roles in various secondary palate subtypes.

Response: We appreciate the reviewer's thoughtful question. Up to this point, there has not been such a single-cell multiome study in the field of secondary palate development, which holds significance not only developmental biology but also human diseases. The novelty of our findings lies in uncovering the specific regulatory mechanisms governing the anterior and posterior trajectories within the secondary palate. While it is true that several transcription factors have been associated with secondary palate development in general, our study delves deeper into elucidating the precise regulatory mechanisms that distinguish the anterior and posterior regions. This provides a more comprehensive understanding of how these transcription factors orchestrate palate development in a trajectory-specific manner.

Minor comments:

1. The authors should color-code or otherwise differentiate the data in Supplementary Fig.1A and B so that readers can assess how well the replicates at each timepoint overlap in each of these metrics. Relatedly, it is difficult to make out the three colors representing the E13.5 replicates in Supplementary Fig.1E. Further, in Supplementary Fig.1D, E13.5 replicates 1 and 2 appear more similar to E14.0 replicate 2 than E13.5 replicate 3. The authors should comment on this discrepancy.

Response: We thank the reviewer for carefully reading our manuscript and this valuable suggestion. We have updated Supplementary Fig.1a, b, e to color-code the replicates at each time point. For Supplementary Fig.1d, the correlation was calculated using pseudobulk values of SCT transformed gene expression, which may induce bias due to cell type proportion differences between replicates. In the embedding plot (Supplementary Fig.1e), we do see the cells from E13.5 replicates 1 and 2 appear more similar to E13.5 replicates 3, while E14.0 replicate 2 are close to the cells from E14.0 replicate 1. We have removed original Supplementary Fig.1d panel in the revised version.

Please also note that we changed the subfigure labels from capital letters to lower case letters during revision to align with the journal's style.

2. Page 4, line 29 and page 5, lines 1-2 refer to Supplementary Fig.1C top and bottom, when the panels are presented as left and right.

Response: We thank the reviewer for pointing this out. We have corrected the main text.

3. The label "Glial" is missing from the y-axis in Figure 1E.

Response: We thank the reviewer for pointing this out. We have added the missing label.

4. The RNA and motif p-values for Twist2 on page 7 do not match the values in Table 1.

Response: We thank the reviewer for pointing this out. After reviewing the code, we found the calculation was performed on sub-samples for the purpose of computational efficiency. As a result, the numbers vary slightly. We have fixed the discrepancy to make them consistent.

5. It is unclear what the top and bottom panels represent at each location in Figure 3D. On page 8, lines 15-16 the authors state that "Inhba was restricted to beneath the epithelial layer in the anterior region", when a similar expression pattern is also noted in the middle region. Similarly, the authors state that "Sim2 and Trps1 were expressed only in the anterior half of the posterior region", though they appear to be expressed medially.

Response: We thank the reviewer for pointing this out. We revised the manuscript based on the reviewer's comments. On page 9, we wrote "The expression of *Cyp26b1* was restricted to the

anterior region and *Inhba* was restricted to beneath the epithelial layer in the anterior and middle region. In contrast, *Meox2*, *Prickle1*, and *Efnb2* were mainly expressed in the posterior region of the palate. Interestingly, *Sim2* and *Trps1* were expressed medially in the anterior half of the posterior region of the developing secondary palate.”

6. On page 8, the authors should define how the “anterior” and “posterior” regions of the secondary palate were delineated for bulk RNA-seq.

Response: We thank the reviewer for pointing this out. In the revised manuscript, we have addressed this by providing a clearer explanation of how we delineated the "anterior" and "posterior" regions of the secondary palate for bulk RNA-seq analysis.

In the revised manuscript, on page 8, we have updated the relevant sentence “To validate the gene signatures of anterior and posterior subpopulations, we performed bulk RNA-sequencing (RNA-seq) after isolating RNA from the microdissected anterior 1/3 (n=3) and posterior 1/3 regions (n=3) of the developing secondary palate at E14.0”.

On page 18, we made the following revision: “The anterior (n=3) and posterior (n=3) 1/3 palatal shelves of E14.0 C57BL/6J mice were microdissected, isolated, and then subjected to bulk RNA sequencing.”

7. The authors should comment on why the differentially expressed genes with the largest fold changes in bulk RNA-seq in Figure 3F were often not detected in the scRNA-seq. Relatedly, it is unclear to this reviewer what is being represented in Figure 3H. Does this indicate that only ~6 transcripts commonly mapped to anterior and posterior locations between datasets?

Response: We thank the reviewer for pointing this out. We examined the differentially expressed genes with large fold changes in bulk RNA-seq but were not detected in the scRNA-seq. We found the sequencing depth of these genes is significantly lower than the random gene set of the same length ($p = 9.4e-13$). In the revised manuscript, we added a new Supplementary Figure 6 and on page 8, we wrote “Although the highly significant correspondence between bulk and single-cell levels, several differentially expressed genes with large fold changes in bulk RNA-seq were not detected in the scRNA-seq data. These genes tended to be expressed at low levels in the bulk RNA-seq, suggesting that the limited sensitivity of lowly expressed transcripts in the scRNA-seq assay may obscure the signals (**Supplementary Fig. 6**)”.

Supplementary Fig. 6. Genes differentially expressed in bulk RNA-seq but undetected in scRNA-seq are lowly expressed. The box plot illustrates sequencing depth (y-axis) across different gene sets (x-axis). The p-value is indicated on the plot.

We apologize for not being clear in Figure 3h. Those 6 points actually are not transcripts. Each represents a bulk RNA-seq sample. In the plot, we treated each bulk RNA-seq sample as a single cell and compared it with each single cell in scRNA-seq data. To avoid confusion, we have removed this panel from the revised version.

8. It is unclear why none of the labeled driver genes for posterior trajectory in Figure 5C appear in Figure 4L, especially *Meox2*. The colors in the legend of Figure 5C do not appear to match the colors in the graph.

Response: We thank the reviewer for this valuable question. Figure 5c presents all driver gene lists (gene adjusted p-value < 0.05 & gene fate correlation >0) while Figure 4l shows refined lists of final lineage-determining TFs using additional criteria of motif enrichment test results (gene adjusted p-value < 0.05 & gene fate correlation >0 & motif adjusted p-value <0.05 & motif fold change >0). *Meox2* did not appear in Figure 4l due to its absence in the JASPAR 2020 database for *Mus musculus* (https://jaspar.genereg.net/search?q=Meox2&collection=all&tax_group=all&tax_id=all&type=all&classes=all&family=all&version=all).

We thank the reviewer for this question and for considering limited number of available position weight matrices for *Mus musculus*, we decided to incorporate additional motif information from all vertebrates. As a result, MEOX2 motif was enriched and the data was included in the updated Figure 4l, which was Figure 5b in the revised version.

On pages 11 and 12, we wrote “MEOX2 showed potential regulatory roles in the middle (motif adjusted p-value = 0.024, motif fold change = 2.87) and end of the posterior trajectory (motif adjusted p-value = 1.85×10^{-3} , motif fold change = 2.06, **Figure 5b**).”

On page 19, we wrote “A total of 746 position weight matrices were loaded from the JASPAR 2020 database using getMatrixSet function in TFBSTools package (version 1.32.0, collection = "CORE", tax_group = 'vertebrates')”.

Regarding the color in the legend of Figure 5c, also identified as Supplementary Figure 12 in the revised manuscript, we thank the reviewer for catching it up. We have fixed it in the revised version.

Response to Reviewer #3

The manuscript provides a comprehensive analysis of the developmental programs of the mouse secondary palate, utilizing single-cell multiome data to simultaneously profile RNA-seq and ATAC-seq in the same cells. The study identified distinct cell types and developmental trajectories, as well as key regulators and signaling pathways involved in the development of the mouse secondary palate. Overall, I find the paper to be a useful reference and data resource for studying developmental processes at single-cell resolution.

- The cell clustering and annotation appear to be driven solely by signals in RNA data, while recent computational methods (e.g. Seurat v4) can handle paired RNA and ATAC data to create joint embeddings of the two modalities for clustering. Therefore, using information from both modalities for clustering would be more reasonable.

Response: We thank the reviewer for the valuable suggestion. To respond to the reviewer's point, we applied Seurat v4 weighted nearest neighbor (WNN) approach and generated a joint embedding, which is depicted in new Supplementary Figure 2a.

In the revised manuscript, on page 5, we wrote "Unsupervised dimension reduction based on either gene expression (scRNA-seq), chromatin accessibility (scATAC-seq) profiles, or both modalities, consistently revealed similar structures as visualized using Uniform Manifold Approximation and Projection (UMAP) (Figure 1b, Supplementary Fig.2a)".

Supplementary Fig. 2. Embedding and cell type annotation is robust across modalities and highly variable gene counts. a UMAP visualization based on different modalities. **b** ...

- It would be helpful to understand how the authors selected the 3000 highly variable genes and whether the annotation results are stable to the chosen genes.

Response: We thank the reviewer for this important question. We used SCTransform for normalizing RNA-seq data normalization, which returns 3,000 highly variable genes by default. According to the tutorial provided by the authors of Seurat (https://satijalab.org/seurat/articles/sctransform_vignette.html), SCTransform effectively normalizes

the data and removes technical effects, thereby enabling the detection of more subtle biological fluctuations by including additional features.

To address the reviewer's concern, we applied multiple cutoffs for highly variable genes, including 1,000, 1,500, 2,000, and 2,500, and performed dimension reduction and clustering analysis. Cell types were annotated independently based on canonical marker expression. The embedding of different iterations shown in new Supplementary Figure 2b, indicating that the annotation results are robust across various cutoffs for the number of highly variable genes.

In the revised manuscript, on page 5, we added “The annotation results are demonstrated to be robust across various cutoffs of the number of highly variable genes (**Supplementary Fig. 2b**).”

Supplementary Fig. 2. Embedding and cell type annotation is robust across modalities and highly variable gene counts. ... b UMAP visualization based on different number of highly variable genes. Cell types were annotated independently based on canonical marker expression. The annotation results are robust across various cutoffs of the number of highly variable genes.

- Checking motif enrichment of ATAC data would be beneficial in addition to checking the enrichment of gene activity scores for differentially expressed genes in the annotated cell types.

Response: We thank the reviewer for the valuable suggestion. When checking motif enrichment of ATAC data, we found a limited number of available position weight matrices for *Mus musculus* in the JASPAR 2020 database. Thus, we updated the results by incorporating additional motif information from all vertebrates. Figure below depicts the motif activity score for each major cell type.

- How consistent is the diffusion pseudotime against real time (days) in the trajectory analysis?

Response: We thank the reviewer for this important question. In the revised version, we added a new Supplementary figure panel depicting the relation between diffusion pseudotime and real time. The increasing trend over time shows the consistency between these two variables.

In the revised manuscript, on page 10, we added “The cells with large diffusion pseudotime values tended to be derived from late time points (**Supplementary Fig.10**).”

Supplementary Fig. 10. The diffusion pseudotime is consistent with real developmental stage for each trajectory. The boxplot below depicts the diffusion pseudotime (y-axis) and real time (x-axis) for each cell in each trajectory.

- The driver genes were inferred by checking marginal correlation with the fate probability, but it should be acknowledged that not all marginally correlated genes are necessarily drivers. There could be many "passenger" genes correlated with the real driver genes, but counted as driver genes using this method. This limitation should be discussed.

Response: We thank the reviewer for this important point. We completely agreed with reviewer's point that there could be many "passenger" genes correlated with the real driver genes. Additional motif enrichment test was conducted to narrow down the driver gene list. In addition, to better differentiate "passenger" genes from real driver genes, we applied CellOracle and performed *in silico* TF knockout simulation analysis. We also re-analyzed published RNA-seq data of *Shox2* knockout to validate our *in silico* perturbation predictions in the revision. We will try to develop independent computational approaches to distinguish the potential driver genes from the passenger genes in future.

We added the following section to the Results section of the revised manuscript on pages 12 and 13:

"*In silico* perturbation analysis analysis finds driver genes

Next, we applied CellOracle to assess the impact of perturbing specific regulators on the development of the secondary palate. This algorithm leverages single-cell multi-omics data to deduce gene-regulatory networks. It then conducts *in silico* perturbations to simulate how these changes affect cellular development, relying solely on unperturbed data.

Our analysis focused on the cells within the anterior and posterior trajectories. Independent data analysis was conducted, including normalization, clustering, and dimension reduction using PAGA and force-directed graphs, followed by diffusion pseudotime calculation. The manifold revealed two distinct trajectories originating from the multipotent cells towards the anterior and posterior cells (**Figure 6a**). We then calculated perturbation scores for all detected TFs. High perturbation scores indicate that *in silico* knockout of the TF significantly decreased the development of the trajectory, suggesting that the TF is an essential driver of the trajectory. Interestingly, while the CellOracle perturbation scores were correlated for many TFs, SHOX2 and MEOX2 showed relatively high specificity for the anterior and posterior trajectories, respectively (**Figure 6b**). Indeed, *in silico* perturbation of *Shox2* and *Meox2* reversed the developmental velocities for the anterior and posterior trajectories, respectively (**Figure 6c, d, Supplementary Fig. 13**).

The regulatory networks predicted by CellOracle for all transcription factors can be found in **Supplementary Table 3**. For SHOX2 and MEOX2, CellOracle predicted 11 and 4 target genes, respectively. It is noteworthy that among these predicted targets, *Satb2*, *Prrx1*, and *Prickle1* have previously been linked to cleft palate (**Figure 6e**).

To validate the predicted gene-regulatory dynamics, we re-analyzed published bulk RNA-seq data derived from the E14.5 anterior hard palatal tissues of *Shox2^{Cre/-}* mice, wherein the *Shox2* gene had been knocked out. Among the 11 predicted *Shox2* targets, 8 genes exhibited significantly altered expression following *Shox2* knockout (adjusted p-value < 0.05) (**Figure 6f**). As expected, genes predicted to be positively regulated by *Shox2* demonstrated decreased expression upon *Shox2* knockout (**Figure 6e, f**). Correspondingly, genes predicted to be negatively regulated displayed increased expression upon *Shox2* knockout (**Figure 6e, f**). Taken together these results suggest that SHOX2 and MEOX2 serve as crucial regulators driving the development of the anterior and posterior secondary palate, respectively."

Figure 6. *In silico* perturbation analysis reveals SHOX2 and MEOX2 as drivers of the anterior and posterior trajectories, respectively. **a** Left: UMAP visualization of CNC-derived mesenchymal subpopulations. Right: Force directed graph visualizations of anterior and posterior subpopulations. **b** Scatter plot shows *in silico* TF perturbation scores for the anterior (x-axis) and posterior (y-axis) trajectories, respectively. **c** Left: CellOracle vector field graphic shows the developmental flow of anterior trajectory. Arrows start from multipotent cells and point towards the anterior subpopulation. Right: CellOracle vector field graphic shows simulated vector shift after *in silico* Shox2 knockout. The developmental flow towards anterior cells is reversed upon *in silico* knockout of Shox2. **d** Same visualization as in panel c, showing simulated Meox2 perturbation in the posterior trajectory. **e** Network graphs visualized using Cytoscape show predicted SHOX2 and MEOX2 regulatory networks. The shape of the arrow heads is based on the predicted direction of regulation. Target nodes are colored based on regulation in *Shox2*-knockout RNA-seq data. **f** Heatmap shows log₂ change of RNA expression in three *Shox2* knockout samples normalized to controls. The bulk RNA-seq data used here was downloaded from Xu et al. 2019.

Supplementary Fig. 13. Shox2 and Meox2 exhibit high perturbation score after *in silico* knockout simulation for anterior and posterior trajectories, respectively. **a** Left: UMAP visualization of data subset showing *Shox2* expression in anterior subpopulation. Middle: Grid plot showing inner product of *Shox2* perturb simulation and anterior trajectory developmental flow. Right: scatter plot shows inner product score (y-axis) along pseudotime (x-axis). **b** Same Visualization as panel A for *Meox2* in the posterior trajectory.

REVIEWER COMMENTS

Reviewer #1 (Remarks to the Author):

The authors improved the manuscript by implementing the suggested changes. I am happy with the final results.

Reviewer #2 (Remarks to the Author):

The authors have addressed several of my previous concerns, through new *in silico* and ChIP-seq experiments and clarifications in the text, which has strengthened the manuscript. However, two of my major previous points remain unaddressed. Importantly, given the demonstrated roles of *Shox2* and *Meox2* in secondary palate development in mouse models, it remains unclear what novel findings are provided in this manuscript that justify publication in *Nature Communications*.

1. The authors still have not discussed which steps of secondary palate development they were capturing at each timepoint. This will be crucial to make the findings clear to a broad audience interested in tissue morphogenesis. To be clear, the authors should be discussing milestones such as downgrowth, elevation, fusion, etc.

2. While the authors have performed *in silico* perturbation analysis for all transcription factors in the anterior and posterior trajectories, *Shox2* and *Meox2* did not have the highest and/or most specific perturbation scores. Further, the predicted *Meox2* regulatory network is relatively small and does not include any genes previously associated with cleft palate. The authors performed ChIP-seq, which indicated enriched *Shox2* and *Meox2* motifs (though the statistical significance of the enrichment is not presented) in the anterior and posterior trajectories, respectively, and demonstrated a peak in one target transcript per transcription factor. However, their conclusions that “*in silico* perturbation analysis identified transcription factors *SHOX2* and *MEOX2* as drivers of the development of the anterior and posterior palate, respectively” and “Cell fate analysis unveils lineage-determining regulators” (among others) is not supported by the data. To back up statements in the manuscript about these driver genes, the authors still need to perform knock down of individual transcription factor function *in vivo* and demonstrate that a specific trajectory is affected and/or perform detailed lineage tracing experiments.

Reviewer #3 (Remarks to the Author):

The authors have addressed all my comments and concerns.

Response to Reviewers

MS ID: NCOMMS-23-08175B

MS Title: Single-cell multiomics decodes regulatory programs for mouse secondary palate development

Response to Reviewer #1

The authors improved the manuscript by implementing the suggested changes. I am happy with the final results.

Response: We thank the reviewer #1 for her/his valuable time on evaluating our manuscript and positive comments on our revision.

Response to Reviewer #2

The authors have addressed several of my previous concerns, through new in silico and ChIP-seq experiments and clarifications in the text, which has strengthened the manuscript. However, two of my major previous points remain unaddressed. Importantly, given the demonstrated roles of Shox2 and Meox2 in secondary palate development in mouse models, it remains unclear what novel findings are provided in this manuscript that justify publication in Nature Communications.

Response: We thank the reviewer #2 for carefully evaluating our revision and the valuable advice. While the importance of Shox2 and Meox2 in secondary palate development has been demonstrated before, the precise lineage-specific regulatory networks at single-cell resolution have not yet been described before. In addition, while our manuscript focused on these two exemplary transcription factors, our data contains the predicted regulatory networks for a much larger collection of transcription factors representing a unique resource to the research community. We highlighted these two transcription factors to illustrate the useful information of our data. In the Results section, on page 16, we wrote:

“While our analysis focused on these two important regulators of secondary palate development, our results provide additional information at genome-scale using single cell multiome approach. In addition to Shox2 and Meox2, we provide inferred regulatory networks for several additional transcription factors, which represents a valuable resource to the research community (**Supplementary Table 3**).”

1. The authors still have not discussed which steps of secondary palate development they were capturing at each timepoint. This will be crucial to make the findings clear to a broad audience interested in tissue morphogenesis. To be clear, the authors should be discussing milestones such as downgrowth, elevation, fusion, etc.

Response: We thank the reviewer for this critical point and constructive suggestion. We apologize for any lack of clarity in the previous revision. To clarify which steps of secondary palate development were captured at each timepoint, we added the following text to the second revised manuscript.

In the Methods section, on page 17, we wrote:

“Palatal shelves were isolated from time-mated C57BL/6J mice (000664, Jackson Laboratory) at four distinct developmental timepoints, including E12.5, E13.5, E14.0, and E14.5. Specifically, at E12.5, we primarily captured the early stages of palatal shelves, during which they emerge as outgrowths from the maxillary processes. E13.5 corresponds to a phase characterized by palatal shelf downgrowth towards the tongue. At E14.0, our samples included palatal shelves in the process of

elevation. At this stage, they undergo a significant transformation, transitioning into a more horizontally oriented position above the tongue. Finally, at E14.5, we isolated palatal shelves during the fusion stage, representing the point at which these structures come into contact and ultimately merge along the midline.”

2. While the authors have performed in silico perturbation analysis for all transcription factors in the anterior and posterior trajectories, *Shox2* and *Meox2* did not have the highest and/or most specific perturbation scores. Further, the predicted *Meox2* regulatory network is relatively small and does not include any genes previously associated with cleft palate. The authors performed CHIP-seq, which indicated enriched *Shox2* and *Meox2* motifs (though the statistical significance of the enrichment is not presented) in the anterior and posterior trajectories, respectively, and demonstrated a peak in one target transcript per transcription factor. However, their conclusions that “in silico perturbation analysis identified transcription factors *SHOX2* and *MEOX2* as drivers of the development of the anterior and posterior palate, respectively” and “Cell fate analysis unveils lineage-determining regulators” (among others) is not supported by the data. To back up statements in the manuscript about these driver genes, the authors still need to perform knock down of individual transcription factor function in vivo and demonstrate that a specific trajectory is affected and/or perform detailed lineage tracing experiments.

Response: We thank the reviewer #2 for the constructive feedback. We appreciate the reviewer's observation that *Shox2* and *Meox2* did not obtain the highest or most specific perturbation scores in our CellOracle analysis. To further confirm the importance of *Shox2* and *Meox2*, we applied an additional computational algorithm. On page 14, we wrote:

“To further confirm the relevance of *Shox2* and *Meox2* in secondary palate development, we applied an additional computational algorithm, SCENIC+⁴⁰, to infer major regulators of the developmental trajectory (**Supplementary Fig. 15**). This analysis identified *Meox2* as the top regulon for the posterior trajectory. Due to limited annotation, SCENIC+ does not contain a *Shox2* regulon. However, the top regulon for the anterior trajectory was *Nfia*, which targets *Shox2* based on the Scenicplus annotation. These results further confirm the importance of both *Shox2* and *Meox2* in the anterior and posterior trajectories, respectively.”

We copied this figure below for your convenience.

Supplemental Fig. 15. SCENIC+ identified regulons for anterior and posterior subpopulations, respectively. Scatter plot shows regulon specificity score (RSS, y-axis) of ranked regulons (x-axis) in anterior (left) and posterior subpopulations (right).

Regarding the Meox2 regulatory network, in the previous version, we employed a stringent p-value threshold (p -value < 0.001) to filter the TF-target links, resulting in a relatively small predicted Meox2 regulatory network. In the revised manuscript, we included a larger regulatory network resulting from less stringent, but still significant, thresholding (p -value < 0.01). This expansion enabled the identification of additional Meox2 targets. Importantly, a significant fraction of these targets was previously associated with cleft palate (p -value = 1.01×10^{-4}). The following text was added to the Results section on page 14:

“We expanded the MEOX2 network, by utilizing a less stringent p-value threshold (p -value < 0.01) and identified a total of 39 target genes. Next, we integrated this list of genes with the CleftGeneDB database⁴⁷ which collects curated genes with experimental evidence for relevance in cleft palate. Importantly, 6 of these 39 MEOX2 targets have previously been associated with cleft palate, which represents a significantly larger overlap than expected by chance (Fisher’s Exact test, p -value = 0.0002, odds ratio = 9.2, **Supplementary Fig. 16a**). For instance, *Cacna1d* has been reported to be related to oral cleft phenotype in the GWASdb SNP-Phenotype Associations dataset⁴⁸. Additionally, *Foxp2* is linked to nonsyndromic cleft lip and/or palate through genome-wide linkage analysis⁴⁹. Notably, our MEOX2 CHIP-seq data provided further evidence of MEOX2 binding to promoter regions of these genes (**Supplementary Fig. 16b**).”

Supplemental Fig. 16. Expanded Meox2 network identifies target genes associated with cleft palate. **a** Expanded MEOX2 regulatory network is visualized using Cytoscape (p -value < 0.01). The shape of the arrow indicates the predicted direction of regulation. Color highlights genes previously linked to cleft palate. **b** Integrative Genomics Viewer (IGV) view depicts ChIP-seq binding peak for MEOX2 in the predicted MEOX2 target *Cacna1d* and *Foxp2*.

Following the reviewer's suggestion, we performed additional validation analyses. Towards this aim, we re-analyzed publicly available H3K27 acetylation ChIP-seq data to validate our scATAC trajectories. On page 10, we added:

"To validate the accuracy of the inferred trajectories, we conducted a comprehensive analysis of previously published H3K27 acetylation data obtained from both the anterior and posterior palate³⁷. Leveraging our extensive multiomic datasets, we identified accessibility peaks associated with genes along the anterior and posterior developmental pathways, respectively. We then assessed the probability of observing enriched overlap between the scATAC anterior and posterior peaks with the corresponding anterior and posterior acetylation tracks. The results revealed a statistically significant enrichment for the anterior trajectory (Fisher's Exact test, odds ratio = 1.50, p -value = 4.81×10^{-3} , **Supplementary Fig. 11**). The posterior trajectory also revealed increased enriched overlap, albeit not reaching statistical significance (Fisher's Exact test, odds ratio = 1.34, p -value = 0.13). Taken together, these data confirm the reliability of our inferred trajectories."

Supplemental Fig. 11. The H3K27 acetylation profiles validate the inferred anterior and posterior trajectories. Bar plot shows overlap (y-axis) between the scATAC anterior and posterior peaks and the corresponding anterior and posterior acetylation tracks (x-axis).

Following the reviewer’s suggestion, we performed additional analysis to quantify the statistical significance of the enrichment between the scATAC peaks and the CHIP-seq data. On page 12, we wrote:

“Despite limited statistical power given the use of duplicates, these results showed marginal significance (One-sided t-test, anterior start p -value = 0.055, posterior middle p -value = 0.09). For example, we observed a CHIP-seq binding peak for SHOX2 but not MEOX2 in the predicted SHOX2 target *Nfia* (p -value = 2.29×10^{-4} , signal = 3.01, **Figure 5d**). Simultaneously, a MEOX2 binding peak was observed upstream of the predicted MEOX2 target *Has2* (p -value = 1.53×10^{-4} , signal = 2.82, **Figure 5e**).”

The Figure 5c was also updated.

Figure 5. Key transcription factors driving cells towards anterior and posterior states are validated in CHIP-seq experiments. ...c Left, boxplot shows the likelihood of peaks mapping near genes upregulated at the start of the anterior trajectory (y-axis) for MEOX2 and SHOX2 binding (x-axis). Right, boxplot shows the likelihood of peaks mapping near genes upregulated at the middle of the posterior trajectory (y-axis) for MEOX2 and SHOX2 binding (x-axis).

In addition, we have adjusted the language in the manuscript to temper statements about the roles of Shox2 and Meox2, referring to them as important regulators rather than drivers.

The major innovation of our study is the derivation of high confidence gene regulatory networks important for the development of mouse secondary palate. We have validated our predicted TF-target relationships through (1) newly generated ChIP-seq experiments of SHOX2 and MEOX2 binding as well as (2) published Shox2-knockout RNA-seq and published H3K27 acetylation experiments. While previous studies^{1,2,3,4} have described the relevance of Shox2 and Meox2 in secondary palate biology, to the best of our knowledge our study is the first to derive high-confidence regulatory networks. While our manuscript primarily focuses on Shox2 and Meox2, our dataset includes results for several other genes, making it a valuable resource to the community. We understand the reviewer's point regarding the *in vivo* knockdown experiments, and we agree that such experiments are valuable. However, the scope of the current work limited our ability to conduct these experiments within the timeframe of this study. We will extend this work for future study which will have specific focus on the regulation mechanisms of these transcriptional factors in palate development. We hope that the additional validations, data integration, and analyses we have performed address the reviewer's concerns and demonstrate the significance of our findings in secondary palate development.

We sincerely appreciate the constructive feedback from Reviewer #2, which has greatly contributed to the refinement and clarity of our manuscript. We remain committed to providing robust and valuable insights into the regulatory programs governing secondary palate development.

In the Acknowledgement section, we have acknowledged the valuable suggestions from all three reviewers who helped improve the manuscript.

References

1. Jin JZ, Ding J. Analysis of Meox-2 mutant mice reveals a novel postfusion-based cleft palate. *Dev Dyn* **235**, 539-546 (2006).
2. Hilliard SA, Yu L, Gu S, Zhang Z, Chen YP. Regional regulation of palatal growth and patterning along the anterior-posterior axis in mice. *J Anat* **207**, 655-667 (2005).
3. Yu L, *et al.* Shox2-deficient mice exhibit a rare type of incomplete clefting of the secondary palate. *Development* **132**, 4397-4406 (2005).
4. Xu J, *et al.* Shox2 regulates osteogenic differentiation and pattern formation during hard palate development in mice. *J Biol Chem* **294**, 18294-18305 (2019).

Response to Reviewer #3

The authors have addressed all my comments and concerns.

Response: We thank the reviewer #3 for her/his valuable time on evaluating our manuscript and positive comments on our revision.

REVIEWERS' COMMENTS

Reviewer #2 (Remarks to the Author):

The authors have addressed my previous concerns.